

# Investigating Ground-Level Ozone Pollution in Semi-Arid and Arid Regions of Arizona Using WRF-Chem v4.4 Modeling

Yafang Guo[1], Chayan Roychoudhury[1], Mohammad Amin Mirrezaei[1], Rajesh Kumar[2], Armin Sorooshian[1,3], Avelino F. Arellano[1,3]

[1] *Department of Hydrology and Atmospheric Sciences*, *The University of Arizona, Tucson, AZ, USA*

[2] *Research Applications Laboratory, National Center for Atmospheric Research, Boulder, CO, USA*

[3] *Department of Chemical and Environmental Engineering*, *The University of Arizona, Tucson, AZ, USA*

*Corresponding to*: Yafang Guo (guoy1@arizona.edu)

**Abstract.** Ground-level ozone ($O_3$) pollution is a persistent environmental concern, even in regions that have made efforts to reduce emissions. This study focuses on the state of Arizona, which has experienced elevated $O_3$ concentrations over past decades containing two non-attainment areas designated by the U.S. Environmental Protection Agency. Using the Weather Research and Forecasting with Chemistry (WRF-Chem) model, we examine $O_3$ levels in the semi-arid and arid regions of Arizona. Our analysis focuses on the month of June between 2017 and 2021, a period characterized by high $O_3$ levels before the onset of the North American Monsoon (NAM). Our evaluation of the WRF-Chem model against surface Air Quality System (AQS) observations reveals that the model adeptly captures the diurnal variation of hourly $O_3$ levels and the episodes of $O_3$ exceedance through the maximum daily 8-hour average (MDA8) $O_3$ concentrations. However, the model tends to overestimate surface $NO_2$ concentrations, particularly during nighttime hours. Among the three cities studied, Phoenix (PHX) and Tucson (TUS) exhibit a negative bias in both hourly and MDA8 $O_3$ levels, while Yuma demonstrates a relatively larger positive bias. The simulated mean hourly and MDA8 $O_3$ concentrations in Phoenix are 44.6 and 64.7 parts per billion (ppb), respectively, compared to observed values of 47.5 and 65.7 ppb, resulting in mean negative biases of -2.9 ppb and -1.0 ppb, respectively.





Furthermore, the analysis of the simulated ratio of formaldehyde (HCHO) to $NO_2$ (HCHO/$NO_2$; FNR), reveals interesting insights of the sensitivity of $O_3$ to its precursors. In Phoenix, the FNR varies by a VOC (volatile organic compound)-limited regime in the most populated areas and a transition between VOC-limited and $NO_x$-limited regimes throughout the metro area with an average FNR of 1.15. In conclusion, this study sheds light on the persistent challenge of ground-level $O_3$ pollution in semi-arid and arid regions, using the state of Arizona as a case study.

## 1. Introduction

Ground-level ozone ($O_3$), or tropospheric $O_3$, is a harmful air pollutant that affects human health and plants (Reich, 1987; Lippmann, 1989; Anderson, 2009; Iriti and Faoro, 2009; Wang et al., 2017; Manisalidis et al., 2020). $O_3$ concentrations are affected by meteorological conditions as well as the concentrations of precursors ( Jacob, 2000; Fiore et al., 2002; Vingarzan, 2004; Monks et al., 2015; Wang et al., 2017). Meteorological factors include intensity of solar radiation, temperature, relative humidity, winds, pressure, and boundary layer height (Trainer et al., 2000). The precursors of $O_3$ include nitrogen oxides ($NO_x$) and volatile organic compounds (VOCs). Besides its significant role in forming $O_3$, $NO_x$, particularly $NO_2$, is also an important pollutant mainly emitted by human activities.

With projections indicating the expansion of aridity zones due to climate change in the future (Asadi Zarch et al., 2017; Huang et al., 2017; Achakulwisut et al., 2019; Straffelini and Tarolli, 2023), there is an anticipated rise in $O_3$ levels under more drought and elevated temperature conditions (Achakulwisut et al., 2019), thereby posing potential challenges to overall air quality, vegetation, and public health. In the face with these projections, there is an undeniable sense of urgency in advancing our comprehension of $O_3$ production mechanisms and refining forecasting model skills, especially within urban arid regions. This imperative arises from the acknowledgement that urban areas in arid climates face a distinctive set of challenges marked by exceptionally low precipitation, elevated temperatures, and unique vegetation. Gaining such insights is crucial for generating effective strategies to mitigate the negative impacts on air quality, vegetation, and the health of urban populations in response to shifting climatic conditions.

Because of the Clean Air Act, average $NO_2$ concentrations have decreased substantially in the U.S. since the 1990s (U.S. Environmental Protection Agency [EPA], National Emissions Inventory



(NEI) air pollutant emissions trends data, http://www.epa.gov/ttnchie1/trends/, 2012, hereinafter referred to as EPA, online report, 2012). For example, the annual 98$^{th}$ percentile of daily maximum 1-hour average $NO_2$ was reduced from 42 ppb to 33 ppb with a 21% decrease in the national

average from 2010 to 2022 (EPA, 2023). VOCs in the atmosphere are generally emitted from two major sources: human activity and biogenic volatile organic compounds (BVOCs) produced by plants. In the U.S., VOC emissions data are tracked by the NEI. According to NEI data, in Maricopa County, where the city of Phoenix resides, total estimated VOC emissions from anthropogenic sources, excluding forest wildfires and prescribed burns, decreased by 35% between

2008 and 2020 (from 0.19 million tons to 0.13 million tons). Most anthropogenic emissions reductions were observed among on-road mobile sources and other industrial processes. As a result, $O_3$ levels have substantially decreased across much of the U.S. (Cooper et al., 2012; Parrish et al., 2022). In 2015, the U.S. EPA lowered the $O_3$ National Ambient Air Quality Standard (NAAQS) to 70 parts per billion (ppb). The design value is defined as the annual fourth-highest

maximum daily 8-h average (MDA8) $O_3$ concentration, averaged over three years. Any area that does not meet this standard is designated as a nonattainment area (NAA). Despite the nationwide decrease of $O_3$ precursors and $O_3$ concentrations, there are still areas where $O_3$ levels exceeded the 2015 NAAQS standard of 70 ppb in 2017 (U.S. EPA Green Book 8-h Ozone 2015). Therefore, for these areas, it is critical to have a detailed understanding of the chemical and meteorological

processes influencing $O_3$ formation so that better pollution control can be put in place to reduce $O_3$ levels.

Identifying and quantifying the various sources that contribute to the formation of $O_3$ is challenging due to the complicated nature of atmospheric chemistry and variability of $O_3$ precursors (Trainer et al., 2000; Duan et al., 2008; Odman et al., 2009; Zare et al., 2014; He et al.,

2019; Fang et al., 2021; Yang et al., 2021; Zhan et al., 2023). First, $O_3$ formation is a complex process that involves the interaction of multiple precursor pollutants, such as $NO_x$ and VOCs, under the influence of sunlight. The chemistry behind these reactions can be highly nonlinear and dependent on numerous variables (e.g., temperature, moisture, cloud cover, and solar radiation) (Trainer et al., 2000). This nonlinearity makes it challenging to predict how changes in emissions

will impact $O_3$ concentrations. In addition, $O_3$ is not limited to areas where its precursors are emitted as it can be transported over long distances. This makes it difficult to attribute $O_3$ levels



solely to local sources, as regional and even global factors can influence local concentrations (Vingarzan, 2004; Monks et al., 2015).

In Arizona, the Phoenix-Mesa metropolitan area is currently designated as a moderate NAA for
$O_3$ and has ranked among the top five of most polluted cities for $O_3$ in the recent 5 years (source: https://www.lung.org/research/sota/city-rankings/most-polluted-cities). Another NAA is Yuma County. Unlike Maricopa County, Yuma is a rural region that has a much lower population and emissions. With Yuma being located on the border of Mexico on the south/southwest and California on the west, its $O_3$ levels thus are significantly impacted by both international and inter-
state transport. Qu et al. (2021) investigated the sources of $O_3$ pollution in Yuma, Arizona, and found strong international influences from Northern Mexico on 12 out of 16 $O_3$ exceedance days. They also performed a sensitivity study with the GEOS-Chem model and found that reducing emissions in Arizona alone would have a minimal impact on mitigating $O_3$ exceedances in Yuma, with only a 0.7% reduction in MDA8 $O_3$. In contrast, reducing emissions in Mexico is estimated
to contribute to an 11% reduction in $O_3$ during these exceedances, bringing MDA8 $O_3$ in Yuma below the standard. Li et al. (2015) applied WRF-Chem with sensitivity experiments and showed that Arizona emissions have a dominant impact on MDA8 $O_3$ concentrations in Phoenix, while southern California's contributions range from a few ppb to over 30 ppb.

While long-range transport of precursors and $O_3$ into Arizona does occur, the primary contributor
to $O_3$ levels remains the in-situ production resulting from local emissions. Because most of Arizona is a semi-arid and arid region with a unique southwest natural environment including weather, climate, and desert plants, it is important to understand how the extreme heat, low moisture, and year-around desert shrubs contribute to $O_3$ production in order to minimize $O_3$ exceedances and improve air quality forecasting (Sorooshian et al., 2023). Additionally, even though Arizona is a
typical desert weather region with high temperatures and low moisture year-round, during the North American Monsoon (NAM) the primary wind flow in Arizona shifts from westerly/southwesterly to southerly/southeasterly, resulting in elevated moisture from the Pacific Ocean and the Gulf of California. Furthermore, unlike the other $O_3$ polluted regions in the Eastern US, which are mainly forest ecosystems, most of Arizona experiences little precipitation—less
than 25 centimeters or 25 to 50 centimeters of rain per year (Paul et al., 2002). The BVOCs are also quite unique in the arid climate region. Geron et al. (2006) found out that in the Mojave and



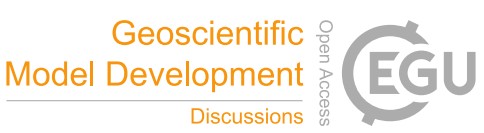

Sonoran Desert regions of the western US where Arizona is, of all the 13 common desert plant species, only two of the species emitted isoprene (most abundant BVOC) indicating that this type of ecosystem is not likely a strong source of isoprene, compared to forest ecosystems.

In section 2 we first discuss the climatology of Phoenix, as a representation of southwest Arizona, and then describe the datasets employed and the setup of the WRF-Chem model. In section 3 we present analyses of model evaluation with observations including meteorological fields, $O_3$, and precursors. The analyses of $O_3$ exceedance and VOC-NOx sensitivity are also included. Section 4 summarizes the main conclusions of this study.

**2. Data and Method**

**2.1 Description of study region and time period**

This research focuses on the study of $O_3$ in the state of Arizona in the U.S. The climate of the south and southwest parts of Arizona (Sonoran Desert) is dry and hot, with much of the region characterized as arid. Our primary interest is in three major cities: Phoenix, Tucson, and Yuma.

Phoenix, the most populated city, is designated as an $O_3$ NAA by EPA along with the entire metro area; Tucson, which is the second largest city in the state, experiences mild $O_3$ pollution but gets stronger influence from the monsoon and Mexico; Yuma, situated near both California and Mexico, is a representation of an arid section of the Sonoran Desert and also designated as a NAA with clean data determination by EPA.

Shown in Figure 1 are the monthly mean surface air values of MDA8 $O_3$, CO, $NO_2$, relative humidity, temperature, and meteorological fields of wind speed, and wind direction in the city of Phoenix. These monthly values were derived from averaging the daily EPA AQS data collected over a 5-year period from 2017 to 2022 at the Phoenix JLG Supersite. The MDA8 (Figure 1a) exhibits peaks during the summer months, spanning from April to September, except for the year

2020 when the COVID-19 pandemic began. On the other hand, the monthly CO, $NO_2$, and relative humidity (RH) show an opposite trend, with their lowest values observed during the summer months. RH is the lowest in June and then increases as the monsoon arrives in July, followed by decreases in September after the monsoon ends. Besides the COVID-19 factor, 2020 is ranked as the second driest year in Arizona's history, with a statewide precipitation level of only 6.63 inches

(NWS Phoenix, 2020). Figure 1(d) shows that the RH levels during late 2020 (red line) and early



2021 (purple line) were the lowest across the five-year period. Additionally, the temperature during the summer of 2020 (Figure 1e) was also the highest. For winds, the windiest seasons are spring and summer, and the wind direction varies throughout the year. Summer months are mostly westerly winds and winter months consist of more easterly winds (Figure 1f-1g). Shown in Figure

1h is the distribution of monthly $O_3$ exceedance days at the JLG supersite in Phoenix (site number: 04-013-9997). An $O_3$ exceedance day occurs when the MDA8 $O_3$ is greater than 70 ppb on that day. The exceedance days are mostly recorded from April to September, referred to here as the "ozone season". In the months of June and July in the year 2020, the MDA8 $O_3$ (Figure 1a) and exceedance days (Figure 1h) were substantially lower than in other years and the reason could be

related to the COVID-19 pandemic. The pandemic's stay-at-home period resulted in much lower traffic levels and hence reduced anthropogenic emissions.

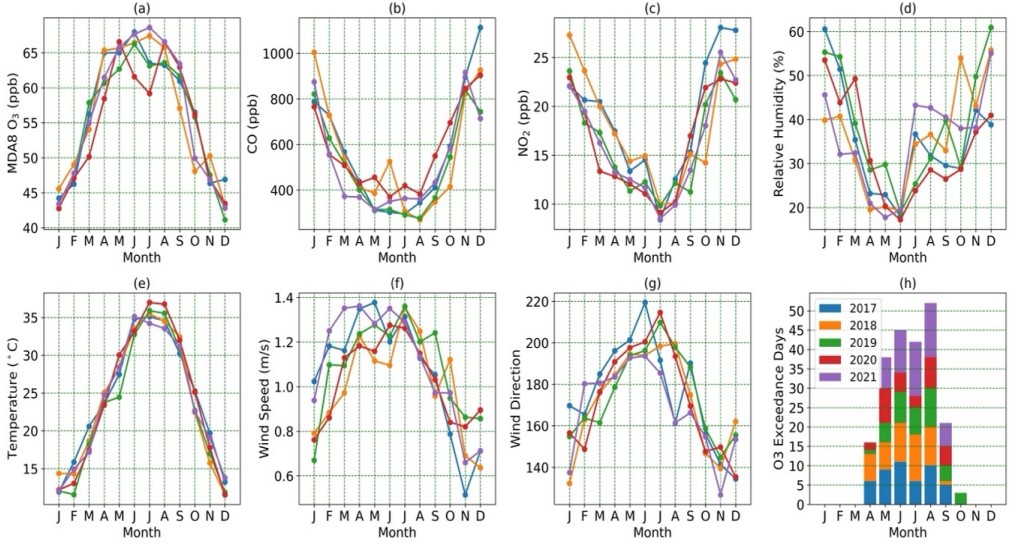

**Figure 1. Monthly mean of Phoenix surface (a) MDA8 $O_3$, (b) CO, (c) $NO_2$, (d) relative humidity, (e) temperature, (f) wind speed, (g) wind direction, and (f) number of exceedance days for years between 2017 and 2021, derived from EPA criteria gases and meteorological daily summary data of a single site (Phoenix JLG supersite).**

Based on these monthly results, we choose the month of June (dry summer), when $O_3$ levels, temperature, and winds are high, and the moisture level is still low. It is also intended to mitigate

the impact of the heavy precipitation that typically accompanies the monsoon. We apply the WRF-



Chem model (v4.4) with state-of-art configurations to simulate the $O_3$ concentrations over Arizona. Numerical simulations were conducted during June between 2017 and 2021 for a total of five years. Furthermore, the ozone season in 2017 was also simulated as our base year. The following sections describe the datasets analyzed herein and the configuration used for the WRF-Chem simulations.

## 2.2 WRF-Chem setup

The Weather Research Forecasting coupled with Chemistry (WRF-Chem) (Grell et al., 2005) model is a fully coupled meteorology-chemistry transport model developed by the National Center for Atmospheric Research (NCAR). This study uses WRF-Chem v4.4 to simulate $O_3$ in Arizona. With our ultimate goal of establishing an operational forecasting and analysis system for Arizona in the future, we have configured the model using the NCAR WRF-Chem forecasting system as a reference (https://www.acom.ucar.edu/firex-aq/forecast.shtml). The comprehensive parametrization schemes are provided in the following list. The Model for Ozone and Related Chemical Tracers (MOZART-4, (Emmons et al., 2010)) is selected for the gas-phase chemistry, coupled with the Goddard Chemistry Aerosol Radiation and Transport (GOCART, (Chin et al., 2002)) for aerosol chemistry with wet scavenging enabled. The standard MOZART-4 mechanism includes 85 gas-phase species, 12 bulk aerosol compounds, 39 photolysis and 157 gas-phase reactions. It also includes an updated isoprene oxidation scheme and a better treatment of volatile organic compounds, with three lumped species to represent alkanes and alkenes with four or more carbon atoms and aromatic compounds (called BIGALK, BIGENE and TOLUENE) (Emmons et al., 2010). The new updated TUV photolysis option, based on standalone TUV version 5.3, is employed to calculate the photolysis rates. This new TUV option uses $O_3$ climatology distributed from the model top (~20km) to 50 km. Initial and lateral boundary conditions are supplied every six hours from both the Global Forecast System (GFS) with a horizontal grid spacing of 1° for meteorology and the Community Atmosphere Model with Chemistry (CAM-Chem) (Lamarque et al., 2012; Tilmes et al., 2015) for chemistry. Biogenic emissions are calculated online with the Model of Emissions of Gases and Aerosols from Nature (MEGAN, v2.1) using the simulated meteorological conditions while running WRF-Chem ( Guenther et al., 2006; Guenther, 2007). Note that MEGAN v2.1 currently is only compatible with the CLM4 (Community Land Model Version 4, Oleson et al. 2010) land surface model. The anthropogenic emissions used in this study





are obtained from 2017 National Emissions Inventories (NEI2017) data provided by the US EPA (https://www.epa.gov/air-emissions-inventories/2017-national-emissions-inventory-nei-data) with a 4 km grid resolution covering the US and surrounding land areas. Biomass burning emissions are calculated using the Fire Inventory from NCAR (FINNv2.5) (Wiedinmyer et al., 2023) and the online plume-rise model (Freitas et al., 2007). FINNv2.5 is based on fire counts derived from both satellite MODIS and VIIRS (Visible Infrared Imaging Radiometer Suite) active fire detection (Wiedinmyer et al., 2023). The following key physics settings are also employed: Morrison double–moment microphysics (Morrison et al., 2009), RRTMG for long and short-wave radiation (Iacono et al., 2008), Eta Similarity for surface layer physics (Monin and Obukhov, 1954), the Unified Noah Land Surface Model (Tewari et al., 2004), the Yonsei University (YSU) planetary boundary layer (PBL) scheme (Hong, 2010), and the Grell–Freitas cumulus parameterization scheme (Grell and Freitas, 2014).

The model is configured with two nested grid domains consisting of 9 km and 3 km horizontal grid spacing along with 34 vertical levels. Shown in Figure 3 is the WRF-Chem domain setup. The parent domain (D01) covers the entire western U.S. with expansion to northern Mexico to better understand the wind shift from Mexico during NAM, while the nested domain (D02, Fig. 2b) focuses on Arizona. Both domains are centered in the Phoenix metropolitan area. D01 features 271 and 394 horizontal grids, while D02 is characterized by 349 and 313 horizontal grids. The topography in Figure 2b (color contours) shows that Phoenix is located in about the center of a valley, called Salt River Valley. The WRF-Chem run periods are specifically designed to be the month of June between 2017 and 2021, with each run consisting of a total of 33 simulation days, including a three-day spin-up in late May and 29 days in June.





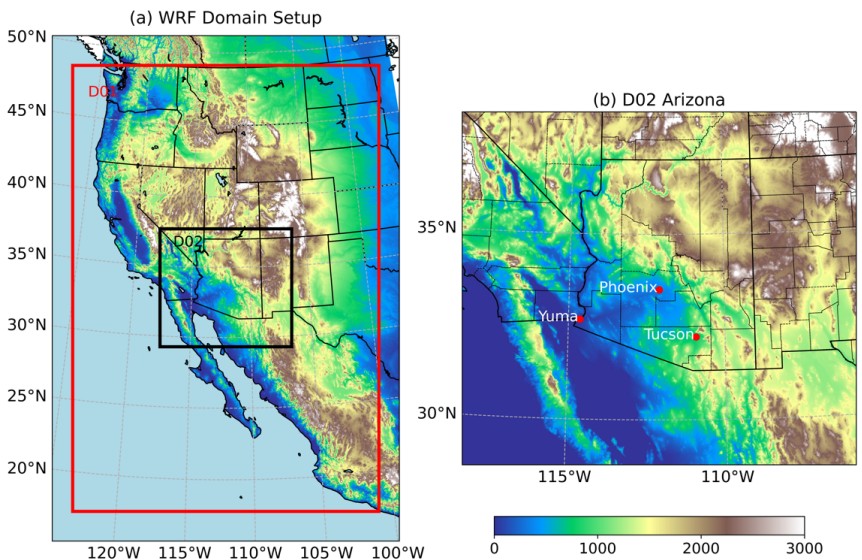

**Figure 2. (a) WRF-Chem domain setup for outer domain D01 and inner domain D02, (b) geographic location of three Arizona cities: Phoenix, Tucson, and Yuma. Black dash lines in (b) represent the county borders. Contours denote the elevation in meters over the continent.**

### 2.3 EPA AQS surface observations

We use the hourly and daily surface in situ observations of $O_3$, CO, $NO_2$, and meteorological fields such as temperature, relative humidity, and winds from the EPA AQS monitoring network (Demerjian, 2000). Sites within each city were selected based on their availability during the study periods for each parameter. For instance, for $O_3$ measurements, 10 sites were selected in Phoenix, 7 in Tucson, and 2 in Yuma. For evaluation purposes, we applied quality control to the raw data

before doing further analysis. To calculate the MDA8 $O_3$, any days with more than 8 continuous hourly data points missing were excluded from the analysis. Zero and negative values were treated as missing while values below the method detection limit (MDL) were replaced with 0.5×MDL.

### 2.4 EPA PAMS VOC measurements

The network of Photochemical Assessment Monitoring Stations (PAMS) established by the U.S.

EPA plays a crucial role in monitoring and understanding ground-level $O_3$ pollution in affected areas providing measurements of various $O_3$ precursors, including VOCs. The list of measurements includes 63 different compounds with some of the most common VOC species like





formaldehyde (HCHO), acetaldehyde, acetone, ethanol, and two monoterpenes (α-pinene and β-pinene; $C_{10}H_{16}$). The primary objective of these PAMS sites is to create a comprehensive database

of $O_3$ precursors and meteorological conditions to better understand local $O_3$ formation, support the development of $O_3$ models, and allow for the tracking of important trends in $O_3$ precursor concentrations over time. The two PAMS monitor sites in Arizona are located in Phoenix (JLG Supersite: 04-013-9997) and Tucson (22nd & Craycroft: 04-019-1011). The sampling frequency for most VOCs is hourly averaged. For formaldehyde, JLG supersite uses the EPA's 3-day

schedule with three 8-hour averaged carbonyl samples per day on every third day.

## 2.5 Radiosonde data

High vertical resolution temperature profiles from radiosondes are applied to determine the planetary layer boundary height (PBLH) for WRF-Chem evaluations. Data from radiosondes launched at three different locations (Phoenix, Tucson, and Yuma) were downloaded. The

radiosonde launches in Phoenix are active during the monsoon season, starting in mid-June and ending in late September while Tucson and Yuma conduct regular daily balloon launches. The launch times for Phoenix and Tucson are set at 0000 UT and 1200 UT, while Yuma operates two launch sites with schedules at 1200 UT, 1800 UT, and 2100 UT. To estimate the PBLH, we use the Bulk Richardson Number Method. Richardson number is a dimensionless number used to

assess atmospheric stability. The top of planetary layer boundary is marked by when the Richardson number exceeds a threshold of 0.25.

## 2.6 CMAQ reanalysis

A high-resolution (12 x 12 $km^2$) air quality reanalysis over the contiguous U.S. (CONUS) is available from 2005-2018 (https://www.gcseglobal.org/development-air-quality-products). This

reanalysis is generated using a newly developed chemical data assimilation system that simultaneously assimilates aerosol optical depth (AOD) retrievals from the Moderate Resolution Imaging Spectroradiometer (MODIS) and carbon monoxide (CO) retrievals from the Measurement of Pollution in the Troposphere (MOPITT) in the Community Multiscale Air Quality (CMAQ) model. The WRF model provides meteorological input for CMAQ simulations over the

CONUS. This dataset offers a suite of air quality products, e.g. $PM_{2.5}$, $PM_{10}$, $O_3$, $NO_2$.

## 2.7 ADEQ forecasts



The Arizona Department of Environmental Quality (ADEQ) produces five-day hourly air quality forecasts for locations across Arizona (https://www.azdeq.gov/forecast). Specifically for our study region, forecasts are released Monday through Friday and include $O_3$, $PM_{10}$, and $PM_{2.5}$. The

forecast values are for the monitor with the highest MDA8 $O_3$ concentration for a given day within the Phoenix-Mesa NAA and the Tucson area, whereas for Yuma it is a single monitor (Yuma Supersite).

## 3. Results and Discussion

### 3.1 Model evaluations

We begin by evaluating the simulated diurnal and monthly variations of meteorological fields and major air pollutants using the AQS monitor site and PAMS observations. Shown in Figure 3 is the time series of Phoenix hourly surface $O_3$ concentrations in June for the year 2017 and 2018. CMAQ air quality reanalysis datasets are also included for evaluation. The diurnal pattern of $O_3$ concentrations is clearly discernible, with peak levels occurring during the afternoon and reaching

their lowest points at night. In general, the WRF-Chem model effectively captures these daily $O_3$ concentration patterns. Conversely, the reanalysis dataset notably underestimates $O_3$ levels during the nighttime. Notably, in June 2017, an extreme $O_3$ event occurred, characterized by $O_3$ levels exceeding 80 ppb and lasting for 9 days, starting on 14 June 2017. On 20 June, $O_3$ levels even reached 100 ppb. The model effectively simulates this exceptional event, while the reanalysis

dataset tends to overestimate this peak.

Listed in Table 1 are the statistical metrics comparing hourly concentrations from the WRF-Chem to the AQS monitoring sites at three different locations: Phoenix (PHX), Tucson (TUS), and Yuma (YUMA). The statistics include Pearson correlation coefficient (R); mean bias (MB); mean error (ME); root mean square error (RMSE); normalized mean bias (NMB); normalized mean error

(NME); mean normalized bias (MNB); mean normalized error (MNE); fractional bias (MFB); fractional error (MFE). For hourly $O_3$, the correlation (R) indicates that all locations show a positive correlation, with PHX having the highest at 0.81, followed by TUS at 0.73, and YUMA at 0.69. The negative MB suggests that in PHX (-2.9 ppb) and TUS (-1.7 ppb) WRF-Chem underestimates the $O_3$ concentration, while YUMA (5.2 ppb) suggests an overestimate. PHX and

TUS generally exhibit smaller biases and errors compared to YUMA. Additionally, YUMA has



the highest variability in errors and the highest NME and RMSE values, indicating less agreement with AQS data compared to PHX and TUS.

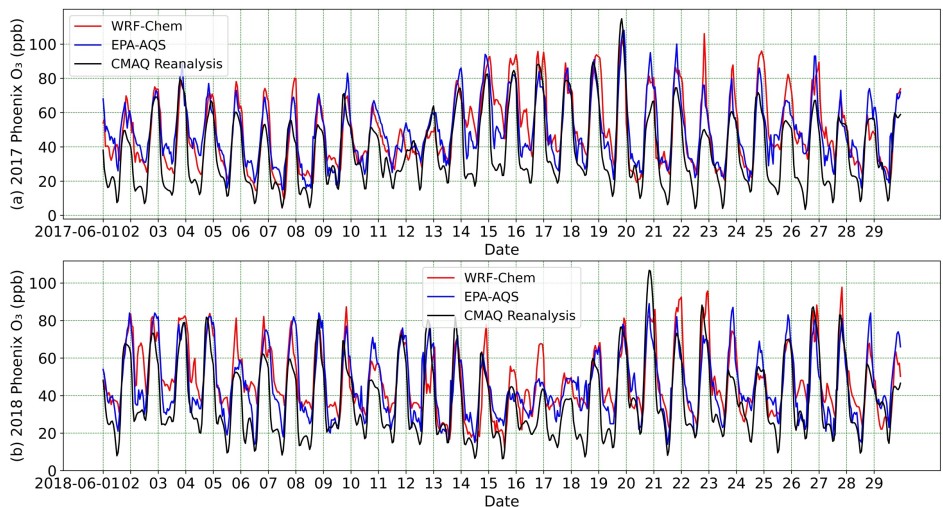

**Figure 3.** WRF-Chem simulated (red), EPA AQS (blue), and CMAQ reanalysis (black) hourly surface $O_3$ concentrations in June of 2017 (top) and 2018 (bottom) in Phoenix. Results are at Universal Time.

**Table 1. Statistics of WRF-Chem hourly $O_3$ and evaluation with respect to EPA AQS observations. R: Pearson correlation coefficient; MB: mean bias; ME: mean error; RMSE: root mean square error; NMB: normalized mean bias; NME: normalized mean error; MNB: mean normalized bias; MNE: mean normalized error; MFB: fractional bias; MFE: fractional error.**

| Hourly $O_3$ | WRF, AQS | R | MB | ME | RMSE | NMB (100%) | NME (100%) | MNB | MNE | MFB | MFE |
|---|---|---|---|---|---|---|---|---|---|---|---|
| PHX | 44.6, 47.5 | 0.81 | -2.9 | 8.3 | 10.6 | -6.1 | 17.6 | -0.03 | 0.19 | -0.07 | 0.20 |
| TUS | 46.2, 47.9 | 0.73 | -1.7 | 6.4 | 8.1 | -3.5 | 13.4 | -0.02 | 0.14 | -0.04 | -0.37 |
| YUMA | 46.3, 41.1 | 0.69 | 5.2 | 9.1 | 12.6 | 12.9 | 22.4 | 0.26 | 0.34 | 0.13 | 0.61 |

In addition to the hourly $O_3$ evaluation, we have also examined the MDA8 (Maximum Daily 8-Hour Average) $O_3$. MDA8 $O_3$ is a crucial metric used in air quality management and assessment, as well as a good indicator of air pollution. Shown in Figure 4 are the MDA8 $O_3$ concentrations for June 2017-2021 in the cities of PHX, TUS, and YUMA. Same as the hourly $O_3$ in Figure 3, for the MDA8 $O_3$, we employed the CMAQ reanalysis data and AQS observations for our evaluation.



Additionally, since the CMAQ reanalysis data is available only up to the year 2018, we incorporated ADEQ forecasts for the years 2019 through 2021. The statistical results of the MDA8 $O_3$ evaluation against AQS observations can be found in Table 2. Statistics of CMAQ reanalysis and ADEQ forecasts in each individual year is included in Supplement Table S2.

Overall, WRF-Chem MDA8 $O_3$ exhibits a smaller mean bias compared to hourly $O_3$, except Yuma,
where the mean bias slightly increases from 5.2 ppb to 6.3 ppb. However, it is worth noting that the correlation coefficients show a slight decrease from 0.81 and 0.73 to 0.66 and 0.62 for PHX and TUS, respectively, compared to hourly $O_3$. This reduction in correlation could be attributed to fewer data points available for linear fitting in the case of MDA8 $O_3$. Additionally, the RMSE at PHX is reduced from 10.6 ppb for hourly $O_3$ to 8.6 ppb for MDA8 $O_3$. Considering statistics in
both Tables 1 and 2, we conclude that WRF-Chem exhibits better performance in capturing the variations of MDA8 $O_3$ concentrations than hourly $O_3$.

Furthermore, when we compare WRF-Chem with CMAQ reanalysis, our findings indicate that WRF-Chem demonstrates smaller biases and higher correlations. For instance, the reanalysis consistently underestimates the MDA8 $O_3$ at PHX but overestimates them at Yuma during the 4-
9 June and 20-28 June periods, as illustrated in Figure 4.

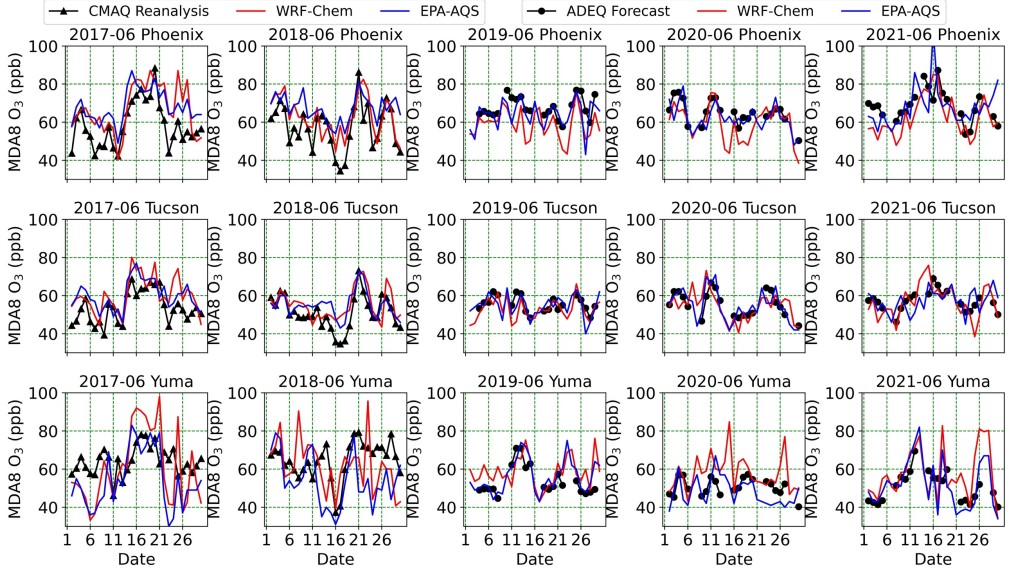

**Figure 4. WRF-Chem simulated (red), EPA AQS (blue), and CMAQ reanalysis (black triangle), ADEQ forecasts (black circles) MDA8 $O_3$ concentrations in June 2017-2021 for three major Arizona cities: Phoenix, Tucson, and Yuma.**





**Table 2. Same as Table 1, but for MDA8 ozone evaluation.**

| MDA8 O$_3$ | WRF, AQS | R | MB | ME | RMSE | NMB (100%) | NME (100%) | MNB | MNE | MFB | MFE |
|---|---|---|---|---|---|---|---|---|---|---|---|
| PHX | 64.7, 65.7 | 0.66 | -1.0 | 6.9 | 8.6 | -5.6 | 10.5 | -0.05 | 0.11 | -0.06 | 0.11 |
| TUS | 55.9, 56.3 | 0.62 | -0.4 | 5.1 | 6.3 | -0.8 | 9.1 | -0.00 | 0.09 | -0.01 | 0.09 |
| YUMA | 59.9, 52.9 | 0.7 | 6.3 | 8.7 | 11.3 | 12.2 | 16.5 | 0.14 | 0.18 | 0.11 | 0.15 |

Besides O$_3$ evaluations, we examined other air pollutants and essential meteorological parameters. We present in Figure 5 the daily surface concentrations of CO, NO$_2$, isoprene, and formaldehyde (HCHO), along with surface temperature (T) and relative humidity (RH) for June 2021. CO and NO$_2$ are two prominent anthropogenic pollutants and serve as O$_3$ precursors. Isoprene and monoterpene are two types of BVOCs that account for 81% of emitted BVOCs. Their concentrations are significantly influenced by factors such as temperature, vegetation, and light conditions (Morrison et al., 2016; Kalogridis et al., 2014). It is important to note that observations of VOCs using the PAMS system, in comparison to the well-established AQS monitoring system, remain relatively limited. Currently, the PAMS monitoring network in Arizona only operates during summer months from June to August and only started in recent years. For instance, of the two PAMS sites within Arizona, only two daily measurements of formaldehyde were recorded in June 2019 in Phoenix, and the observation schedule changed from 1 in 6 days to 1 in 3 days since 2018. In Tucson, formaldehyde observations only became available starting in 2021 with a 1 in 3 days schedule. Daily measurements of isoprene became available in both Phoenix and Tucson starting in 2021.

In comparison with the observations, the model appropriately replicated the daily variations of surface T and RH with minimal biases. However, for CO, WRF-Chem failed to capture the elevated episode over PHX during 11-15 June 2021. It is worth noting that during this period there was an active wildfire (Telegraph Fire, situated southeast of Phoenix, https://wfca.com/wildfire-articles/arizona-fire-season/) that lasted one month and became one of the largest wildfires in the U.S. throughout the 2021 wildfire season. Because of this, the CO levels in both Phoenix and Tucson were significantly impacted by the fire plumes with smoke moving right over Phoenix.



The model may not be able to simulate the smoke plumes well. Despite the limited PAMS data, we were able to compare the daily isoprene concentrations with observations in both cities. On average, daily mean isoprene is around 5 ppb in PHX and 1 ppb in TUS. Furthermore, for HCHO concentrations, the model is comparable to the observations, not only in terms of the values, but also in capturing their variations. In conclusion, the online biogenic emission model employed in

the WRF-Chem model, MEGAN 2.1, effectively simulates the BVOC levels.

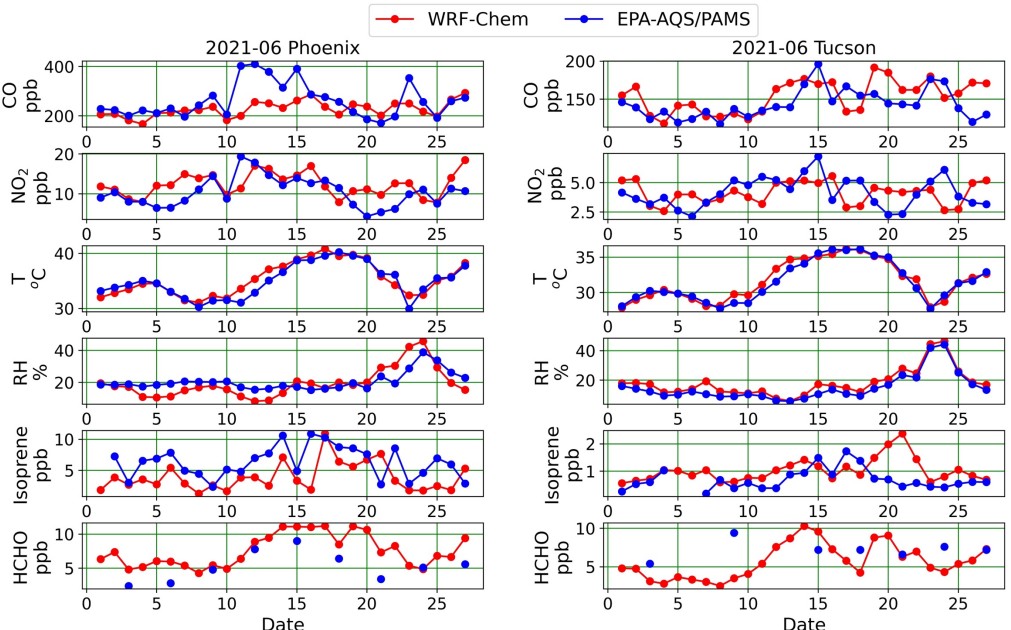

**Figure 5. WRF-Chem simulated (red) and EPA observed (blue) surface concentrations of CO and NO₂, surface temperature (T), 2-meter relative humidity (RH), surface concentrations of isoprene and formaldehyde (HCHO) for June 2021. CO, NO2, T, and RH measurements are obtained through the EPA AQS network. Isoprene and HCHO measurements are acquired from the EPA PAMS networks in PHX and TUS.**

Figure 6a-6c presents the spatial plots of monthly mean surface concentrations of MDA8 $O_3$, CO, and NO₂ for June. The contour plots are based on hourly model output between year 2017 and 2021. The colored circles represent the AQS surface observations for three cities: Phoenix (PHX),

Tucson (TUS), and Yuma (YUMA). Both the WRF-Chem model and the observations indicate that MDA8 $O_3$ in the Phoenix metro area reaches up to 65 ppb (Figure 6a). The northeast of PHX, which is a downwind region, experiences significant $O_3$ pollution as the prevailing winds in June





are predominantly southwest winds (Figure 6f). The background $O_3$ level in most of Arizona is around 50 ppb, while west/southwest Arizona, including Yuma, is substantially influenced by $O_3$ from California.

For CO concentrations, the highest simulated surface levels in PHX reach 224.2 ppb, closely matching the corresponding AQS measurement of 221.9 ppb (Figure 6b). Downwind of Phoenix, CO concentrations range between 100 to 120 ppb. Hotspots in the southeast direction of both PHX and TUS are associated with wildfire burning events, such as the 2017 Frye Fire (southeast hotspot of TUS) and the 2021 Telegraph Fire. The observed mean $NO_2$ level in PHX is approximately 5 ppb and is mostly distributed in populated areas as the main source of $NO_2$ is anthropogenic emissions. An additional figure in the supplement (Figure S1) provides monthly mean $O_3$, CO, and $NO_2$ concentrations for individual years.

The aridity of southwest Arizona is characterized by high temperatures and low relative humidity. Shown in Figures 6d-6f are the mean surface temperature, 2-meter relative humidity (RH), and surface winds. Notably, the temperature in PHX is slightly higher than that in TUS as PHX is located in the valley and TUS has a higher elevation (see Figure 2b). The RH overall is under 20% in southwest Arizona, where the Sonoran Desert is located. The climate of the west/southwest and other parts of Arizona is distinctive. The monthly mean wind predominantly comes from the southwest direction, with an average speed of 10 miles per hour (mph).



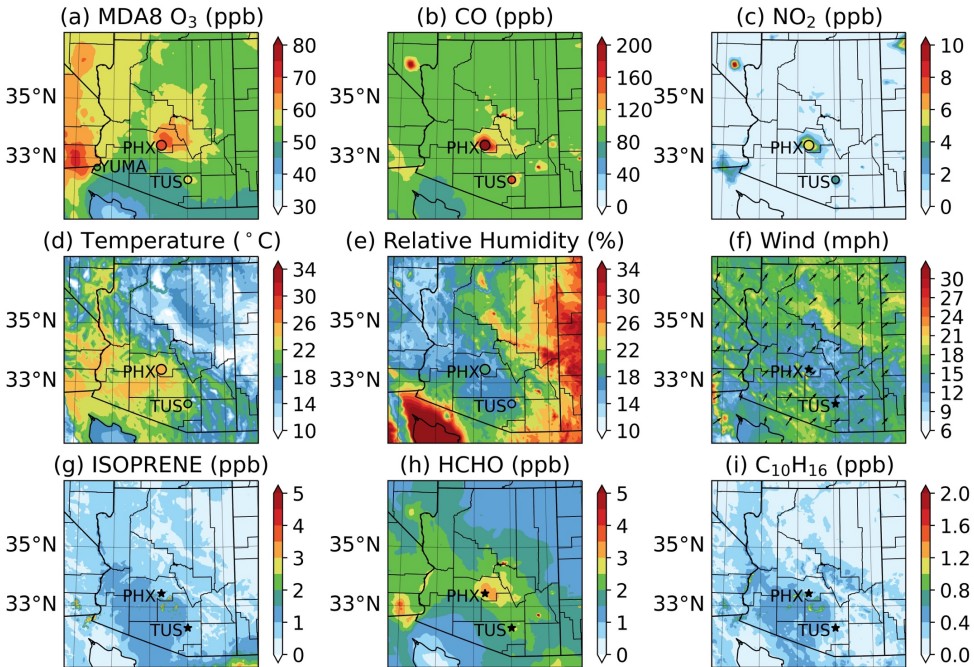

**Figure 6. WRF-Chem simulated monthly mean concentrations of main pollutions (O₃, CO, NO₂), meteorological fields (temperature, relative humidity, wind), and major VOCs (isoprene, formaldehyde, and monoterpene). Colored circles represent the EPA AQS site observations for comparison.**

Table 3 presents the statistics of CO, NO₂, T, and RH between simulations and observations for PHX and TUS. In general, the simulated values of O₃, CO, and T (temperature) align well with the observations. Temperature shows a small, normalized bias of 2% and -1.3% for PHX and TUS, respectively. The model overestimates CO both in PHX and Tucson by 7.1% and 5.75%, respectively. Additionally, the model overestimates the NO₂ levels in both PHX and TUS. Figure 6g-6i also demonstrates three dominant VOC concentrations: isoprene, formaldehyde (HCHO), and monoterpene (C₁₀H₁₆). Overall, the BOVCs are rather small over the desert region, except Yuma, where it is largely impacted by agricultural vegetation.





**Table 3. Statistics of WRF-Chem evaluation with respect to EPA AQS monitors. Results represents the average of June across five years between 2017 and 2021.**

| City | Method | CO (ppb) | NO$_2$ (ppb) | T (°C) | RH (%) |
|---|---|---|---|---|---|
| Phoenix | AQS | 221.8 | 9.0 | 24.8 | 18.4 |
| | WRF | 238.0 | 9.5 | 25.3 | 15.6 |
| | Bias (%) | 16.2 (7.1%) | 0.5 (5.3%) | 0.5 (2%) | -2.8 (-15.2%) |
| Tucson | AQS | 142.1 | 3.9 | 24.0 | 16.3 |
| | WRF | 150.2 | 4.4 | 23.7 | 17.1 |
| | Bias (%) | 8.1 (5.7%) | 0.5 (12.8%) | -0.3 (-1.3%) | 0.8 (4.9%) |

To further investigate the bias between simulations and observations, in Figure 7, we present the frequency distributions of hourly O$_3$, the corresponding O$_3$ bias with respect to the AQS observations, and MDA8 O$_3$ for June in the five-year period for Phoenix (top), Tucson (middle), and Yuma (bottom). For O$_3$ levels higher than 50 ppb (background O$_3$ level in Arizona), WRF-Chem demonstrates good performance in estimating the distributions in Phoenix and Yuma but tends to overestimate in Tucson, particularly between 50 to 60 ppb. Furthermore, WRF-Chem fails to capture the extremely high O$_3$ observational days exceeding 70 ppb for all three cities. Conversely, for low O$_3$ levels below 50 ppb, which are more associated with nighttime O$_3$, WRF-Chem substantially underestimates the values. Therefore, for bias analysis, we divide the assessment into daytime and nighttime periods to account for the diurnal variability of O$_3$ formation. The middle panel in Figure 7 presents the fractional bias of hourly O$_3$ between WRF-Chem and AQS observations. In general, during the daytime the mean bias is positive (Figure 7b, 7e, 7h) suggesting an overestimation by WRF-Chem, while a negative mean bias during the night indicates that WRF-Chem underestimates the hourly O$_3$ values in PHX. The MDA8 O$_3$ distribution demonstrates better overall agreement between the model and observations than hourly O$_3$, consistent with the statistics in Tables 1 and 2.



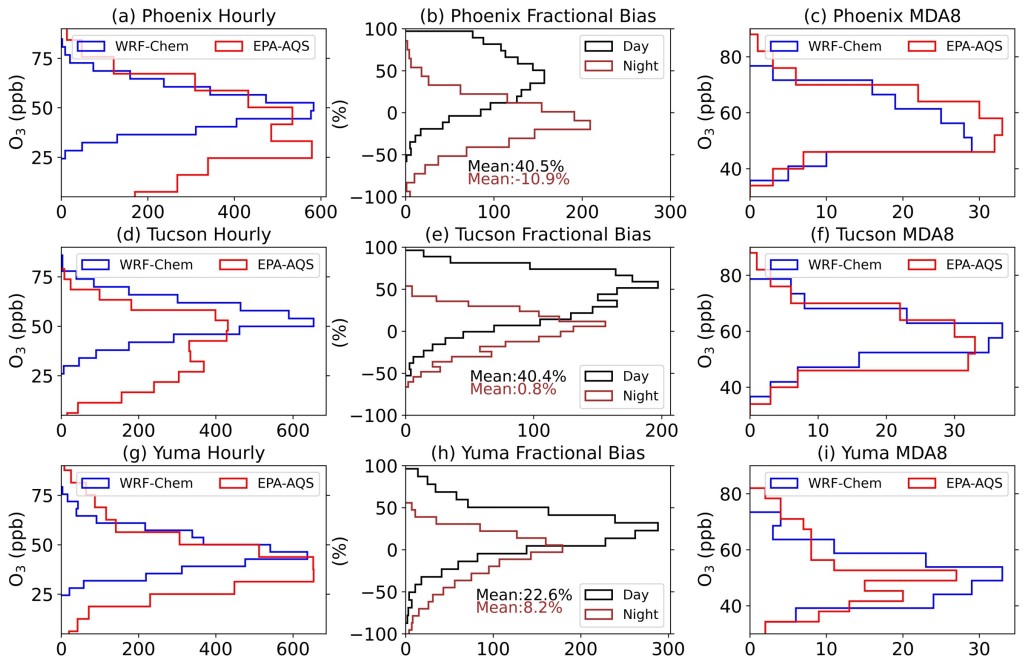

**Figure 7. Model evaluation for cities of Phoenix (top row), Tucson (middle row), and Yuma (bottom row). The first panel in each row shows observed (red) and WRF-Chem simulated (blue) surface $O_3$ frequency distribution; the second panel is the frequency distribution of model bias for both daytime (black) and nighttime(brown); the third panel presents the frequency distribution of MDA8 $O_3$.**

To gain deeper insights into the factors contributing to $O_3$ bias between daytime and nighttime, the distribution of surface $NO_2$ concentration is presented in Figure 8. For data quality purposes, surface $NO_2$ concentrations that are less than 0.5 ppb are discarded for both simulations and observations. Similar to $O_3$, the model misrepresents large $NO_2$ episodes in PHX and TUS when $NO_2$ is greater than 40 ppb and 15 ppb, respectively. There is a larger diurnal variability in the

observations than in simulations. The simulated $NO_2$ distribution during daytime and night are comparable while observed distributions are significantly different with distinct slopes (Figure 8a, 8d). In PHX, the model overestimates the high $NO_2$ levels (>10 ppb) during the night while in TUS, the model underestimates the $NO_2$ during the daytime. The mean bias for day and night in PHX are 0.2 ppb and 1.9 ppb, respectively. The mean bias for day and night in TUS is -2.4 ppb





and 0.5 ppb, respectively. The bias over Tucson suggests that WRF-Chem overestimates the NO$_2$ during the night.

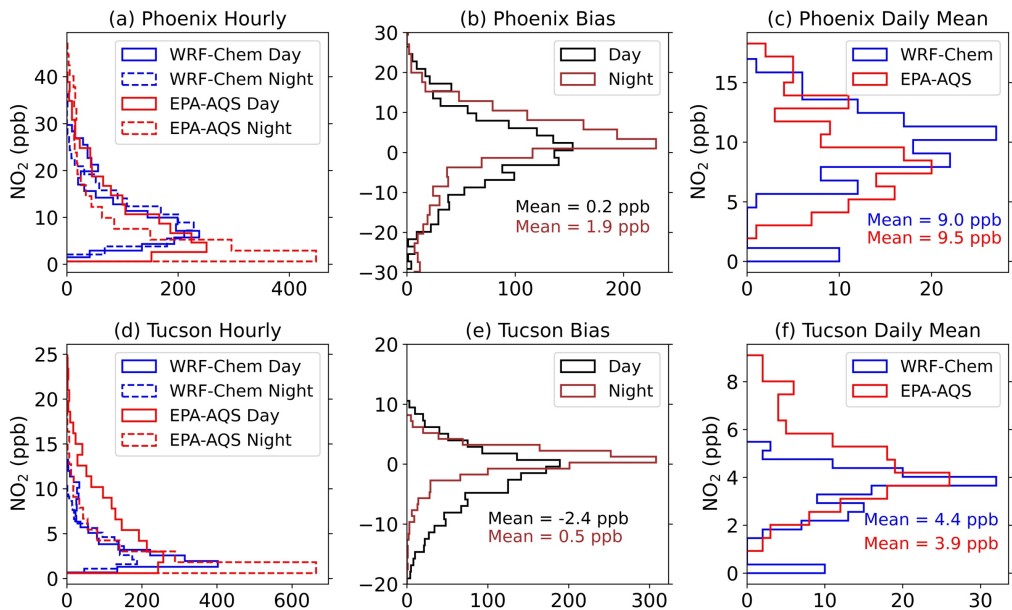

**Figure 8. Same as Figure 7, but for surface NO$_2$ concentrations at two cities: PHX (top) and TUS (bottom). From left to right panels: hourly surface NO$_2$ distributions, NO$_2$ fractional bias, daily mean NO$_2$ distributions. Hourly NO$_2$ distributions on the left panel are divided into day and night times.**

To address the biases depicted in Figure 8 during daytime and nighttime, the PBLH is investigated. A higher PBLH allows pollutants and aerosols to disperse and mix with cleaner air over a larger vertical extent, resulting in a reduction of air pollutant concentrations. Consequently, an overestimation of PBLH leads to an underestimation of O$_3$ and NO$_2$, and conversely, an underestimation of PBLH may contribute to overestimations of these pollutant levels. Using the

radiosonde data, we estimated the PBLH at three cities and compared with model simulations. The launching times of radiosondes at Arizona sites are at Universal Time (UT) hours of 12:00, 21:00, and 00:00, which correspond to Local Time (LT) hours of 05:00, 14:00, and 17:00, respectively.





Presented in Figure 9 are the PBLH for three cities, PHX, TUS, and YUMA, during June 2018 (additional years' data are available in the supplement). The nighttime soundings launched at LT 05:00 are highlighted with red stars, and their corresponding WRF-Chem simulated PBLH values are represented as red dots. Conversely, daytime soundings are indicated by blue markers. Simulated PBLH at all other times without sounding data are labeled with grey dots. It is worth noting that the WRF-Chem model consistently demonstrates an underestimate of PBLH during nighttime (as denoted by the red markers) and an overestimate during daytime (as shown by the blue markers). The mean daytime bias of PBLH between model and observations at Phoenix (LT 17:00), Tucson (LT 14:00), and Yuma (LT 14:00) are 322.0 m, 18.1 m, and 602.5 m, respectively. These biases are closely related to the MDA8 $O_3$ bias listed in Table 2 where bias in Phoenix and Tucson are negative and positive in Yuma. The nighttime biases are all negative with values of -509.7 m, -435.4 m, -55.8 m, indicating an overall underestimate. The underestimate of PBLH during the night will cause the shallower vertical mixing of daytime accumulated $O_3$ leading to the positive bias of nighttime $O_3$ observed in Figure 7.



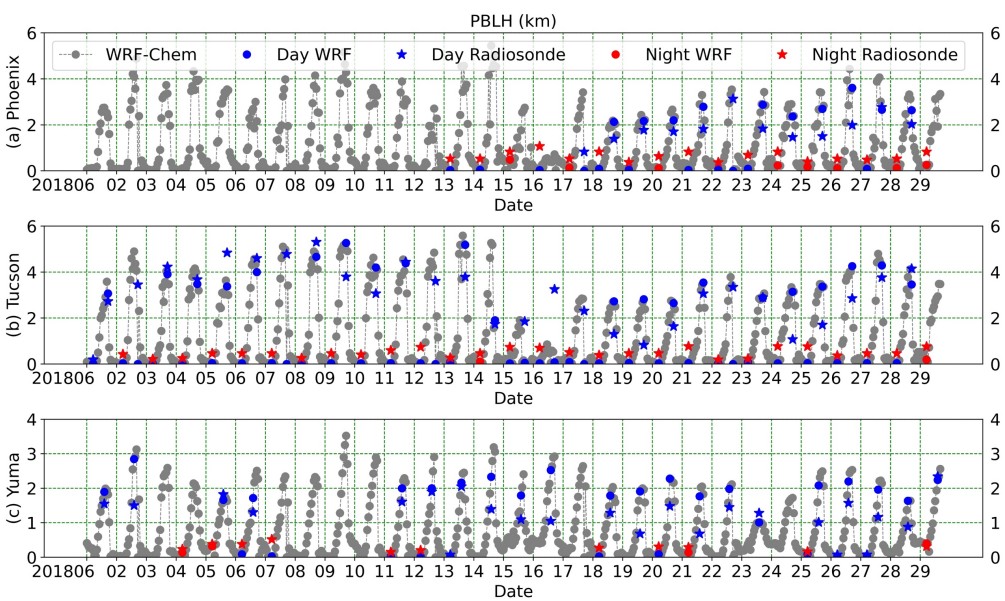

**Figure 9. Planetary boundary layer height (PBLH) in June 2018 for three cities: (a) Phoenix, (b) Tucson, and (c) Yuma. Dots and stars represent simulation from WRF-Chem and observation from radiosondes, respectively. Nighttime PBLH estimated from radiosonde data at 05:00 local time are denoted as red stars, with their corresponding simulated PBLH values indicated as red dots. Blue markers represent radiosondes launched during daytime with corresponding WRF-Chem simulations denoted by blue dots.**

### 3.2 O$_3$ Exceedance

According to the EPA, an exceedance day occurs on each calendar day when the MDA8 O$_3$ concentration is greater than 70 ppb, where 70 ppb is the ground-level O$_3$ standard from the 2015 National Ambient Air Quality Standards (NAAQS). A design value, on the other hand, is a statistic that describes the air quality status of a given location relative to the NAAQS level. The O$_3$ design value of the Phoenix-Mesa metropolitan area has increased from 76 ppb in 2017 to 81 ppb in 2022

(refer to Table S1 in the Supplements). The rising and persisting O$_3$ levels led to the reclassification of Phoenix-Mesa metropolitan area from a marginal to a moderate non-attainment status for O$_3$ limits by the EPA. In the previous section, we demonstrated that WRF-Chem exhibits good performance in simulating the mean O$_3$ and other precursor parameters. Moreover, the model performs better with the MDA8 O$_3$. To further investigate the issue of O$_3$ pollution in Phoenix,

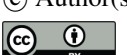



this section focuses on $O_3$ exceedances. As depicted in Figure 1, $O_3$ exceedances typically first start in April and last occur in September, with the exception of the year 2019 when exceedances extended into October, and the year 2021 when no exceedance was observed in April. Over the five-year period from 2017 to 2021, in the greater Phoenix area, the average annual count of $O_3$ exceedance days was 43.4. Even in 2020, amidst the onset of the COVID-19 pandemic and the

enforcement of stay-at-home measures, which resulted in reduced concentrations of $NO_x$, $O_3$ exceedances in Phoenix did not exhibit significant reduction. Figure 10(c) illustrates the boundary of the designated Maricopa County non-attainment area (NAA, depicted by polygons outlined in black), along with the locations of AQS sites equipped for $O_3$ monitoring. In total, there are 29 monitoring sites, with 27 of them situated within the NAA boundary.

Presented in Figure 10 are the spatial variations of the mean $O_3$, MDA8 $O_3$, and count of $O_3$ exceedance days for June 2017 within the Maricopa County NAA. In the top panel (Figures 9a-c), we depict the monthly mean surface hourly $O_3$ concentrations as derived from WRF-Chem, CMAQ reanalysis, and data from AQS monitor sites. These 29 AQS sites, encompassing a range of urbanization levels, population densities, and downwind/upwind positions, exhibit considerable

variability even within the NAA. Figures 10(d-f) show the monthly mean MDA8 $O_3$ concentrations, with higher levels observed in the northeastern part of the NAA, a pattern accurately captured by both WRF-Chem and the reanalysis. Better agreement among model and observations is evident considering both hourly and MDA8 $O_3$. The reanalysis data substantially underestimates the mean $O_3$ levels by 10 ppb but captures the MDA8 $O_3$ spatial distribution

pattern. Lastly, the count of $O_3$ exceedance days is shown in Figure 10(g-i). The exceedance days vary from 2 to 10 days within the area. The regions with the highest population density, particularly the central Phoenix-Mesa region, exhibit the highest counts of exceedance days. In general, the model exhibits strong agreement with the observational data regarding factors such as location, number of days, spatial extent, and spatial variability.

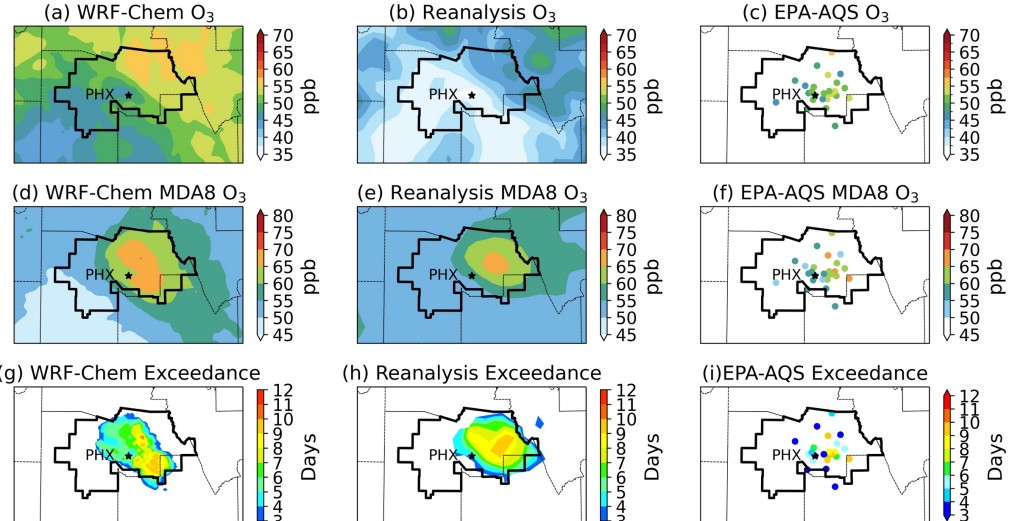

**Figure 10. WRF-Chem simulation (left column), CMAQ reanalysis (middle column), and EPA AQS observations (right column) of monthly mean surface hourly and MDA8 $O_3$ concentrations(top and middle), and exceedance days (bottom) in June 2017. An $O_3$ exceedance is defined as a day with MDA8 exceeding 70 ppb.**


### 3.3 $O_3$ source attribution

Source attribution of $O_3$ is challenging due to the complex processes that control $O_3$ formation. Tropospheric $O_3$ levels are influenced by a multitude of factors including 1) meteorological factors, such as temperature, relative humidity, cloud cover, radiation, wind speed and direction,

precipitation, and boundary layer height; 2) $O_3$ precursors, such as $NO_x$, VOCs, and CO, which can originate from biomass burning (wildfire, prescribed fire), biogenic emissions, and anthropogenic emissions; and 3) $O_3$ transport, such as long-range transport and stratospheric $O_3$ intrusions. Understanding the relationships between these factors and $O_3$ levels is essential for discerning their respective impacts on ambient $O_3$ concentrations. Several analytical methods are

available for investigating $O_3$ source attributions, e.g., backward trajectory analysis (Xiong and Du, 2020; Dimitriou and Kassomenos, 2015; Betito et al., 2023), machine learning algorithm (Cheng et al., 2023; Mishra et al., 2023; Weng et al., 2022), and chemistry models (Butler et al., 2020; Lupaşcu and Butler, 2019; Sudo and Akimoto, 2007). In this paper, we employ scatter plots that utilize both model outputs and ground observations. These scatter plots serve as a practical

means to delve deeper into the intricate connections between $O_3$ and its major influencing factors,





aiding in the identification and quantification of their contributions to $O_3$ concentrations in the atmosphere.

Figure 11 presents a series of scatter plots that illustrate the relationships between $O_3$ concentrations and other key variables, including CO, $NO_2$, surface temperature (T), and relative humidity (RH) during daylight hours at the Phoenix JLG Supersite. The data points are color-coded, with green denoting simulations and orange representing observations. Each column panel within the figure corresponds to the respective month of June for individual years spanning from 2017 to 2021. The displacement between the orange and green dots on the first row suggests that WRF-Chem overestimates the CO concentrations in all the years except 2018. In the years 2017 and 2021, more extreme $O_3$ concentrations were present with levels exceeding 100 ppb.

The negative correlation between $NO_2$ and $O_3$ (depicted in Figure 11f-j) reveals that in Phoenix, surface $O_3$ levels tend to be higher when $NO_2$ concentrations are lower. When hourly $NO_2$ levels exceed 25 ppb, $O_3$ concentrations generally remain below 60 ppb. Furthermore, the positive correlation between temperature and $O_3$ suggests that in general elevated temperatures are associated with higher $O_3$ levels. It is worth noting that on some extreme hot days, $O_3$ levels can also be low. Conversely, the negative correlation between RH and $O_3$ indicates that increased relative humidity tends to be linked with lower $O_3$ concentrations. These intricate relationships offer valuable insights into the complex interplay between $O_3$ and its influencing factors within the Phoenix JLG Supersite region.



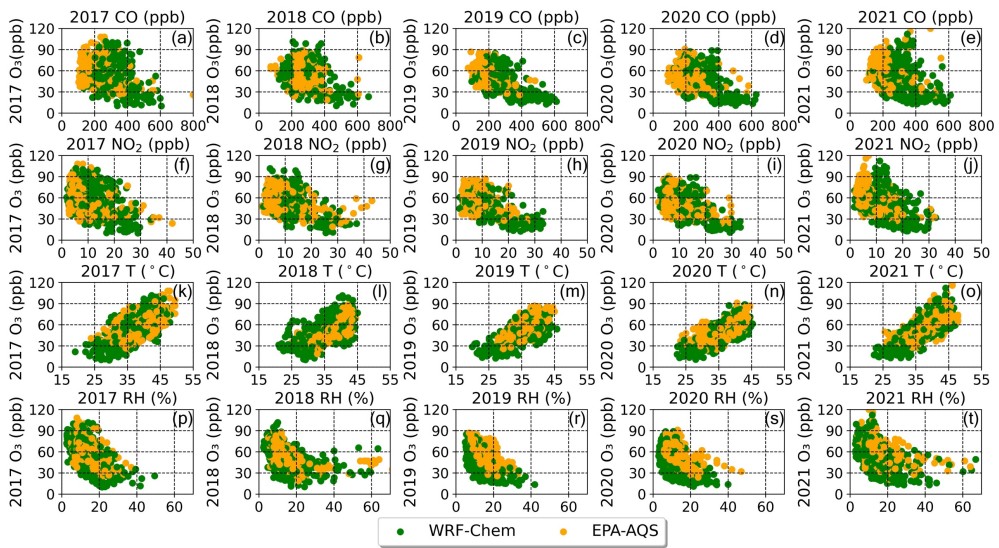

**Figure 11. WRF-Chem simulated (green) and EPA AQS (orange) hourly CO, NO$_2$, surface temperature, and relative humidity versus hourly O$_3$ concentration during the daytime.**


As previously discussed in earlier sections, the O$_3$ exceedance in Arizona can be originated from a combination of various contributing factors, which can be classified into two main categories: local production and transport. Notably, on 13 June 2017, the observed surface O$_3$ levels in both Phoenix and Yuma experienced a substantial increase, with a MDA8 concentration of

approximately 90 ppb in Phoenix. This particular event has been successfully captured by both the WRF-Chem model and CMAQ reanalysis, as illustrated in Figure 4. Shown in Figure 12 are the simulated vertical profiles of O$_3$, CO, NO$_2$, and HCHO, as well as the surface meteorological parameters including PBLH, temperature (T), 500 mb height, and RH during this extreme event. The simulated and AQS observed MDA8 O$_3$ are also included for reference. During the event,

both the surface and columnar concentrations of CO, NO$_2$, and HCHO were all elevated, particularly in the boundary layer. In the meantime, PBLH and RH decreased, while temperature and 500 mb height increased, consistent with the correlation relationships observed in Figure 11. Furthermore, we employed the Hybrid Single-Particle Lagrangian Integrated Trajectory model (HYSPLIT) to calculate back-trajectories for the 13 June exceedance event, as illustrated in

supplementary Figure S5. The obtained trajectories suggest a potential connection between this exceptional event and inter-state transport.



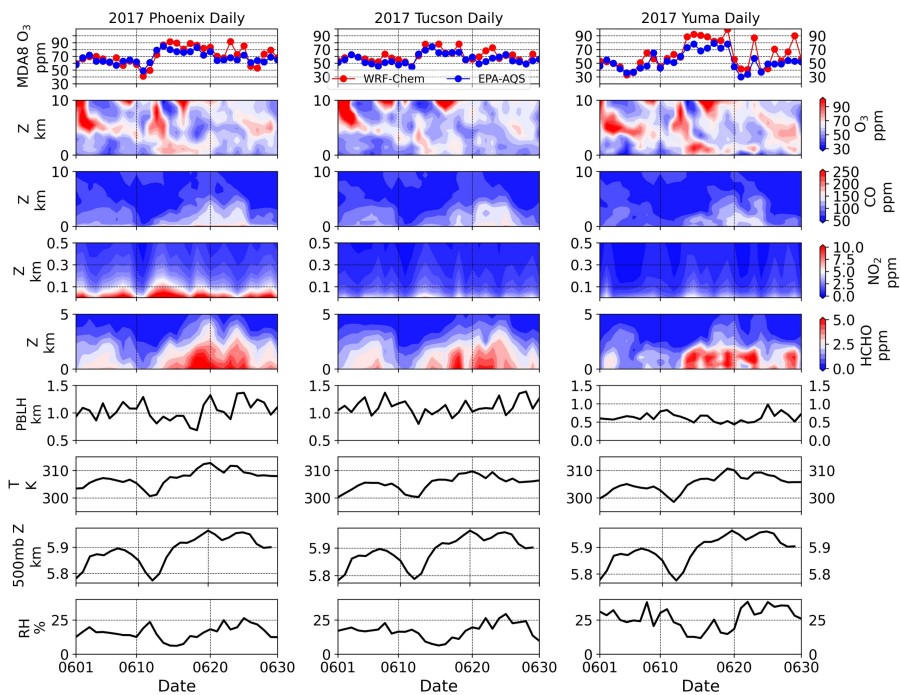

**Figure 12. Vertical profiles of simulated O₃, CO, NO₂, and HCHO at three cities (contour plots) along with the surface MDA8 O₃ (top panel), PBLH, surface temperature, 500 mb height, and surface relative humidity in June 2017. An O₃ exceedance event on 13 June was observed in all three cities.**

### 3.4 O₃-NOₓ-VOC sensitivity

The O₃-NOₓ-VOC sensitivity is a crucial concept in the fields of atmospheric chemistry and air
quality management (Duncan et al., 2010; Sillman, 1995; Sillman and He, 2002; Sillman et al., 2003; Liu and Shi, 2021; Carrillo-Torres et al., 2017; Zaveri et al., 2003). It refers to how the concentration of O₃ in the atmosphere responds to changes in the levels of NOₓ and VOCs. Understanding this sensitivity is essential for assessing and managing air quality, particularly in regions where O₃ pollution is a concern. The sensitivity is often quantified as the ratio of VOC to
NOₓ, an important parameter for characterizing the efficiency of O₃ formation in the environment. When this ratio is high, O₃ formation is constrained primarily by the availability of NOₓ, leading to what is defined as NOₓ-limited or NOₓ-sensitive chemistry. Consequently, taking measures to reduce NOₓ emissions directly correlates with O₃ reduction. Conversely, under lower ratios, it is





referred to as a VOC-limited or VOC-sensitive regime. In these scenarios, $O_3$ levels are notably
more responsive to reductions in VOCs, and solely decreasing $NO_x$ may not effectively lower $O_3$
concentrations and even worse may increase $O_3$ levels. Ratios between the two regimes are
considered transitional, and both $NO_x$ and VOC controls may be effective.

However, it is important to acknowledge that the specific range of ratios used to define VOC and
$NO_x$ limitations can vary among researchers and depend on the specific dataset and variables under
consideration. Different studies and regulatory assessments may employ distinct criteria for
categorizing $O_3$ sensitivity to VOCs and $NO_x$, making it imperative to consider these variations
when interpreting and applying sensitivity analyses in different contexts. From an observational
perspective, the formaldehyde (HCHO) concentration has been widely used as a proxy for VOC
reactivity as it is a short-lived oxidation product of many VOCs and positively correlated with
peroxyl radicals (Sillman, 1995), and it is also available in many observational datasets. Satellite
data, like TROPOMI (The Tropospheric Monitoring Instrument), provides daily columnar HCHO
and $NO_2$ spatial distributions at a certain time of the day. Thus, satellite data have been widely
used in determining the VOC-$NO_x$ sensitivity regimes (Duncan et al., 2010; Souri et al., 2020; Jin
et al., 2017; Martin et al., 2004). In this study, we employ the HCHO-to-$NO_2$ ratio (FNR) as a
proxy for assessing VOC-$NO_x$ sensitivity. When FNR is less than 1, it is classified as VOC-limited;
when it falls between 1 and 2, it is considered a transitional regime; and when FNR exceeds 2, it
is defined as $NO_x$-limited.

In the previous section, we demonstrated a negative correlation between $NO_2$ and $O_3$ indicating
that Phoenix falls within the VOC-limited/VOC-sensitive regime. To gain a more comprehensive
understanding of $NO_x$-VOC sensitivity in the greater Phoenix metropolitan area, we calculated
monthly FNR values for each year and their respective means. Figure 13 displays spatial maps of
FNR across Phoenix and Tucson, highlighting grids with FNR values less than or equal to 4. The
Maricopa County Non-Attainment Area (NAA) is outlined in red. Overall, central Phoenix is
predominantly characterized as VOC-limited or transitional, with FNR values consistently below
2. Additionally, Phoenix exhibits lower FNR values compared to Tucson. Notably, hotspots related
to fire activities are evident in different years, such as the eastern region of Phoenix in 2019, the
northeastern areas of Phoenix and Tucson in 2020, and the eastern part of Phoenix in 2021. The
varying contours from year to year indicate slight differences in sensitivities between those years,
with 2019 and 2020 showing lower mean FNR values over the NAA compared to other years.



In general, central Phoenix is characterized as VOC-limited or transitional, with an average FNR

of 1.15 across the metropolitan area. The FNR tends to be lower, placing it in the VOC-limited

regime, within the more densely populated urban areas. As one moves towards the suburban areas,

there is an increase in FNR, marking a transition from the VOC-limited regime to the boundary

between VOC-limited and NOx-limited conditions. Phoenix has a lower FNR than Tucson with

higher $NO_2$ levels. Hotspots in different years represent fire activities, such as the east of Phoenix

in 2019, northeast of Phoenix and Tucson in 2020, east of Phoenix in 2021. Fire and biomass

burning activities typically result in significant emissions of CO, $CO_2$, $NO_x$, VOCs, particulate

matter, methane, and more. Consequently, when these fire events occur, they can alter the NOx-

VOC sensitivity of the affected areas. The "pop-up" local FNR minima in Figure 13 (labeled as

WF) suggests that wildfire events lead to a reduction in the FNR and a shift in the sensitivity

regime towards VOC-limited. Similar results have been reported using satellite observations (Jin

et al., 2017) and ground-based surface measurements (Miech et al., 2023) where they found that

during the fire event, the $NO_x$ values are high near the fire leading to lower FNRs. Overall, the

area of contours varies year by year indicating the slight differences of sensitivities between years.

In the year 2019 and 2020, the mean FNR over the NAA is smaller than in other years.

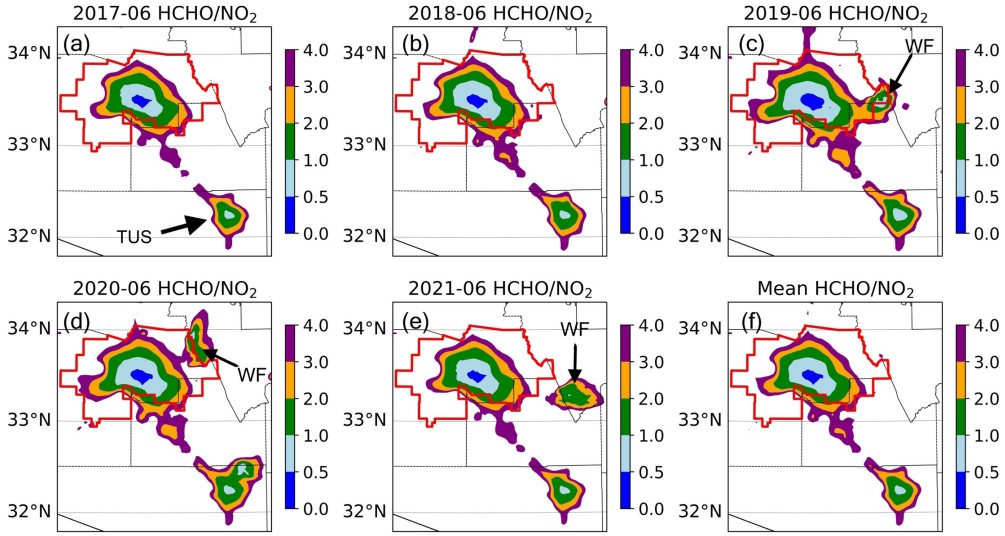

**Figure 13. WRF-Chem simulated monthly mean ratio of surface HCHO/NO₂ over Phoenix and Tucson. Red lines represent the nonattainment area designated by the EPA.**



In Figure 14, we present scatter plots illustrating the relationship between hourly surface concentrations of NO$_2$ and HCHO in three cities: Phoenix, Tucson, and Yuma, as simulated by WRF-Chem. The color gradients in these scatter plots correspond to the respective O$_3$
concentrations (panels a-c) and FNR values (panels d-f).

When we compare Figure 14(a) with 14(d), we observe that in Phoenix, elevated O$_3$ concentrations are linked to lower NO$_2$ levels (as also seen in Figure 11) and high HCHO concentrations, falling within the range of 0.5 to 1.2 in terms of FNR. Conversely, the lowest O$_3$ levels occur when NO$_2$ levels are relatively high. In Tucson, NO$_2$ levels are approximately half of
those observed in Phoenix, and O$_3$ occurrences are less frequent. Higher O$_3$ concentrations in Tucson are primarily associated with FNR values greater than 1. Yuma, on the other hand, exhibits the lowest levels of NO$_2$, but it has the highest HCHO concentrations, also accompanied by high O$_3$ levels. Notably, the mean FNR in Yuma is also the highest among the three cities, as indicated by the prominent red color in Figure 14(f).


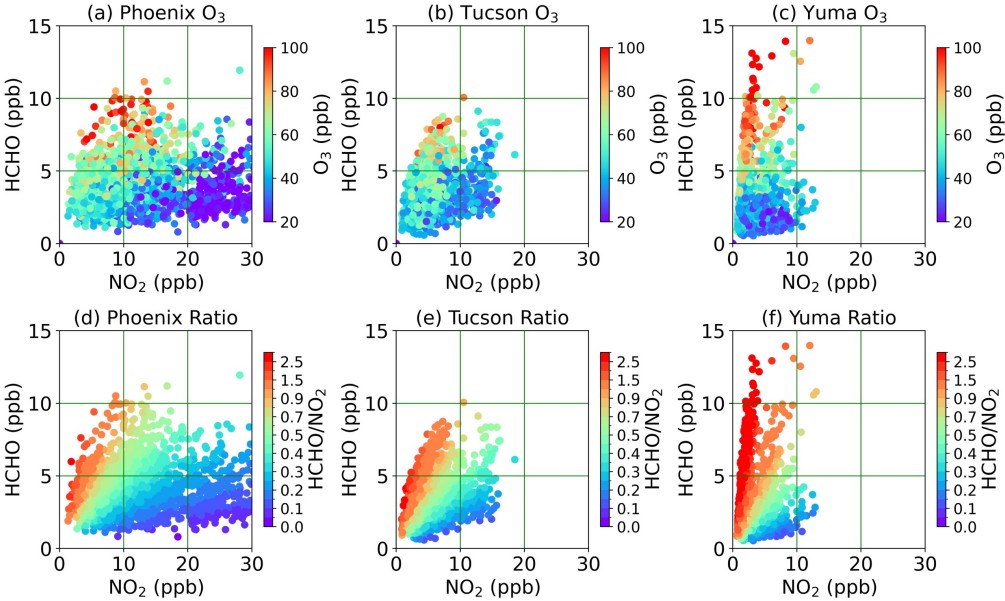

**Figure 14. Scatter plots of WRF-Chem simulated hourly surface NO$_2$ versus HCHO concentrations at three cities: Phoenix, Tucson, Yuma. The colors represent the corresponding O$_3$ concentrations (top) and ratio of HCHO/NO$_2$ (bottom) for years from 2017 to 2021.**





## 4. Conclusion

In this study, our primary objective was to gain a comprehensive understanding of surface $O_3$ pollution in an arid/semi-arid climate region, with a specific focus on the state of Arizona as a representative case study. To achieve this, we employed WRF-Chem simulations to simulate $O_3$ and various other gases, examining the month of June within a five-year period spanning from 2017 to 2021. Our model's performance was assessed by comparison with surface observations from the EPA AQS and PAMS monitoring networks, as well as a CMAQ reanalysis product. Our analysis primarily focused on three major cities within Arizona: Phoenix, Tucson, and Yuma. We calculated statistics for both hourly and MDA8 $O_3$ concentrations. We also examined additional monthly mean fields, including key meteorological parameters (temperature, relative humidity, wind), and air pollutants ($NO_2$, CO, VOCs). Results show that WRF-Chem demonstrated better performance in simulating MDA8 $O_3$ compared to hourly $O_3$. The model exhibited a tendency to overestimate nighttime $NO_2$ levels, resulting in larger biases during the night; the model also shows an overestimate of surface $NO_2$ in Phoenix and an underestimate of $NO_2$ in Tucson. Among the cities examined, Yuma displayed the highest mean error and a positive bias, whereas Phoenix and Tucson showed closer agreement with observations, featuring smaller errors and negative biases. Furthermore, our evaluation indicated minimal biases in the representation of meteorological parameters. However, for VOCs, the model underestimated their surface concentrations.

$O_3$ exceedances were also investigated and evaluated at all available AQS monitoring sites in Maricopa County. Our model exhibited strong agreement with site measurements regarding both the magnitude and the number of days on which an $O_3$ exceedance occurred considering factors such as location, number of days, spatial extent, and spatial variability. The back-trajectory analysis of an $O_3$ exceedance case on 13 June 2017 suggests that Arizona is substantially affected by inter-state transport of $O_3$ from California.

To better understand the $O_3$ formation in this arid/semi-arid region, we examined the correlation between $O_3$ and other factors influencing $O_3$ production. In Phoenix, the scatter plots exhibited negative correlations between $O_3$ and CO, $NO_2$, and RH, while positive correlations were observed with T and HCHO. These correlations strongly suggest that $O_3$ levels are higher when $NO_2$ concentrations are lower and HCHO concentrations are higher, indicating that Phoenix falls within the VOC-limited regime. Additionally, our spatial maps of the FNR confirmed that in the most





densely populated urban areas, the region predominantly falls within the VOC-limited or transitional regime. This analysis significantly contributes to our understanding of $O_3$ dynamics in arid and semi-arid regions and has implications for air quality management and policy in such
environments.

However, a better performance of the model can be pursued through various strategies. While achieving a higher spatial resolution, such as 1 km, is desirable, it remains constrained by the available computational resources and the resolution of input datasets, for instance, the National Emissions Inventory (NEI) currently operating at 4 km resolution. A potential remedy involves
employing a finer-resolution emission dataset, such as the Neighborhood Emission Mapping Operation (NEMO) proposed by Ma and Tong (2022). Additionally, refining simulations of nighttime chemistry, crucial for accurate predictions, necessitates a more precise estimation of the Planetary Boundary Layer Height (PBLH), which, if improved, can contribute to reducing the O3 bias during nocturnal hours. This improvement can be achieved by assimilating PBLH estimates
obtained from radiosonde and ceilometer data.

For future work, we aim to continue investigating the contributions of individual sources of $O_3$ to total $O_3$ levels. We will adopt a tagging technique developed by Emmons et al. (2012) and Butler et al. (2018). This tagging technique uses the WRF-Chem model with the MOZART gas chemistry mechanism to attribute the sources contributing to tropospheric $O_3$. We will focus on the
contributions from anthropogenic, fire, and biogenic emissions, and also use the model to trace the transport of $O_3$ and its precursors ($NO_2$, VOC) from their source.

*Code and Data Availability Statement.*

The WRF-Chem model is version 4.4 is available for download from ZENODO (doi: 10.5281/zenodo.10479471) and publicly available at NCAR https://www2.mmm.ucar.edu/wrf/users/download/get_source.html (last access: 25 June 2022). The model outputs, ADEQ forecast, and CMAQ reanalysis datasets can be provided upon request to the corresponding author. EPA AQS and PAMS hourly and daily datasets are
available at https://aqs.epa.gov/aqsweb/airdata/download_files.html.



*Author contributions.*

YG and AA designed the research. YG performed the model runs and subsequent analysis. YG wrote the paper with contributions from AA and AS. RK provided the reanalysis dataset, AM and CR helped with the observational data acquisition and preprocessing.

*Competing Interests.*

The authors declare that they have no conflict of interest.

*Acknowledgments.*

We especially acknowledge Gabriele Pfister (NCAR/ACOM) and Rajesh Kumar (NCAR/RAL) for kindly providing the NCAR WRF-Chem forecasting and reanalysis system configurations, for which this study is being built upon. We also thank Matthew Pace and Michael Graves at the Arizona Department of Environmental Quality (ADEQ) for providing their forecasts for the state of Arizona.

*Financial Support.*

This work is supported by an Arizona Board of Regents (ABOR) Regent's Grant from the Technology and Research Initiative Fund (TRIF).

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
