# Peer review of "Investigating Ground-Level Ozone Pollution in Semi-Arid and Arid Regions of Arizona Using WRF-Chem v4.4 Modeling"

_Geoscientific Model Development, 2023_

## Author Response (AR1)

Authors' comments in response to all comments made in the open discussion phase.

We thank the three Anonymous Referees for their thoughtful and helpful comments on our submission. We have revised our manuscript taking all these comments into account. Here we repeat the comments in *italic font*, and in each case provide our responses and a summary of the resulting changes to the manuscript (if any) in normal blue font. We also append a copy of the revised version of the manuscript with the changes highlighted at the end of this author's comment.

Reviewer #1

*This is a review for "Investigating Ground-Level Ozone Pollution in Semi-Arid and Arid Regions of Arizona Using WRF-Chem v4.4 Modeling." The co-authors use WRF-Chem to understand and assess ozone concentrations and production chemistry in three cities in Arizona USA. The paper is well-written, the methods are clearly described, and the figures are easy to understand. The analysis includes extensive model evaluation using observations, sondes, and model reanalysis data, and an interesting assessment of FNRs in the region. I believe this paper to be suitable for publication following a few minor revisions as listed below following a brief list of my biggest lingering questions:*

- *Was windblown dust included in the emissions? Would this have an impact on ozone photolysis in AZ?*

Thank you for your insightful question regarding the inclusion of dust emissions and their potential impact on ozone photolysis. In Arizona, dust events are very common, especially in the southwest part where the Sonora Desert is located. According to Lader et al. 2016, the highest frequency of dust storm events happens during the Monsoon season (in July and August).

The presence of dust can have impacts on ozone photolysis dynamics due to its interaction with sunlight. Dust particles can scatter and absorb solar radiation, altering the photolysis rates of ozone molecules in the atmosphere. According to Lader et al. (2016), the highest frequency of dust storm events happens during the Monsoon season (in July and August), and the dust storms occur the most at 6-7 pm when the photolysis rates are lower and hence weaker ozone production.

In our model configuration, we employed the GOCART dust option to account for the dust emissions, also we have chosen June as our main study period to reduce the impacts of dust. We have also included a statement in section 2.1 and in the conclusion section 4.

- *What justification do you have for using the FNR values for theses AZ cities?*

Thank you for pointing out the FNR justifications. There have been different studies looking at the FNR values for the determination of the regimes.

The concentration of formaldehyde (HCHO) serves as an indicator for volatile organic compound (VOC) reactivity as it exhibits a positive correlation with proxy radicals (Sillman,

1995). Sillman (1995) identified that elevated $HCHO/NO_y$ ratios typically indicate $NO_x$-limited regimes, whereas reduced $HCHO/NO_y$ ratios are indicative of VOC-limited regimes. Martin et al. (2004) found that during summer, the transition between radical- and $NO_x$-limited regimes occurs at a particular ratio threshold. Using the Community Multiscale Air Quality (CMAQ) model with finer resolution for the entire continental U.S., Duncan et al. (2010) proposed that formaldehyde-to-nitrogen oxides ratios (FNRs) below 1 suggest a VOC sensitivity regime, FNRs between 1 and 2 indicate a transition zone between VOC and $NO_x$ sensitivities, and FNRs above 2 are characteristic of a $NO_x$-sensitive regime. It is important to note that variations in meteorological variables, emission sources, and pollution levels can alter the ozone production regime. In different studies, various FNR thresholds are calculated. i.e., satellite column retrievals of FNR of 0.7–2.3 in Schroeder et al. (2017), and 3.2–4.1 in Jin et al. (2020). In addition, Acdan et al. (2022) used ground-based PAMS measurements and suggested a FNR of 0.3–1.0 for transition over the Lake Michigan region. In our study, we are following Duncan et al. (2010) which linked FNR with surface $O_3$ sensitivity in model simulation and used in several studies (Tang et al., 2012; Jin and Holloway, 2015, Souri et al., 2017) by defining FNRs less than 1 as VOC sensitivity regime, FNRs between 1 and 2 as a transition between VOC and $NO_x$ sensitivities ('the transitional zone'), and FNRs greater than 2 as $NO_x$-sensitive regime. The definitions are updated in the text in section 3.4.

- *What do your findings in the paragraph at lines 586-94 suggest for regulatory actions to reduce ozone concentrations? E.g. how should the cities determine emissions reductions strategies and under what ozone conditions?*

Thank you for the insightful question. The correlation between HCHO and $NO_2$ levels, and how they correspond to $O_3$ or FNR levels, shows intriguing patterns. Urban areas tend to exhibit higher $NO_2$ levels, resulting in lower FNRs and typically higher $O_3$ concentrations.

Our ongoing investigation into the diurnal cycles and weekend effect of $O_3$ over Arizona has uncovered significant insights. For instance, during the early morning hours over Phoenix, a VOC-limited regime dominates as $NO_2$ levels rise during rush hours. Subsequently, with increasing temperatures, more VOC/BVOCs are emitted, leading to more $O_3$ production (due to increased photolysis rates) and $NO_2$ consumption. Consequently, the FNR rises, transitioning towards a $NO_x$-limited regime.

In Phoenix, the FNR across the Phoenix metropolitan area resides within the transitional regime, and elevated $O_3$ levels often correlate with higher HCHO and lower $NO_2$ levels. The prospect of further reducing $NO_2$ levels may lower $O_3$ design values, but it is possible to increase the daily mean $O_3$ levels due to the intricate interplay of diurnal complex $O_3$ production. In Yuma, the HCHO is higher and $NO_2$ is lower. Elevated $O_3$ is highly correlated to HCHO levels. Therefore, reducing VOC may lower $O_3$ levels.

We have also added a paragraph in section 3.4 regarding the regulatory suggestions from our findings as below:

"Understanding these correlations between HCHO, NO$_2$, and O$_3$ levels is crucial for formulating effective regulatory strategies aimed at mitigating O$_3$ pollution in urban settings, resulted from localized O$_3$ production. The transitional regime observed in the Phoenix metropolitan area suggests that while additional reductions in NO$_2$ levels could potentially decrease O$_3$ design values, there exists the possibility of concurrent increases in daily mean O$_3$ levels due to the intricate interplay of diurnal complex O$_3$ production. In Yuma, where higher HCHO levels prevail, reducing VOC emissions may serve as a viable approach to lowering O$_3$ concentrations."

*Minor suggestions:*

- *Figure 1 caption lists two figure 1f, second should be figure 1h*

Thank you for the comment, we have revised accordingly.

- *What was the CMAQ reanalysis data used for? "For evaluation" Please include a brief description of what CMAQ reanalysis data is used for in the methods.*

Thank you for the suggestion. We have added an extra sentence in section 2.6 to be clearer on the purpose of including this reanalysis dataset.

- *Were fire emissions data inputs year-specific?*

Thank you for the comment. The fire emission is the daily and year-specific high-resolution dataset from satellite fire detections.

- *Line 445: I understand this is an average, but there can't be partial exceedance days. I recommend rounding down.*

Thank you for the feedback. We have changed the number accordingly.

- *There is some duplicate information in the paragraphs starting at line 553 and line 565. For instance, lines 565-6 are a repeat of information in lines 558-60, and the description of the fire hotspots are repeated in the second paragraph. Please clean this up so as not to repeat your findings unnecessarily.*

Thank you so much for pointing this out. We have reorganized the paragraphs and removed repeated sentences.

Review #2

*The authors conducted a multi-year air quality simulation using WRF-Chem to describe and evaluate O3 and its influencing factors in three cities in Arizona, USA. They also analyzed ozone sources and chemistry. The paper is well-written and organized, with a clear introduction to the method, comprehensive use of observational and reanalysis data for evaluation, and thorough*

*presentation of results. This study contributes to improving understanding of O3 formation, transport, and mitigation in arid and semi-arid regions. I am supportive of the theme of this manuscript and believe it can be published after minor revisions.*

1. *The 4 km model-ready emissions from NEI2017 are used while the resolutions of the simulation are 3 km and 9 km. Please clarify how you made the emission regridding. Do you aggregate the elevated point source emissions to the surface layer or allocate them to the model 34 vertical layers?*

   Thank you for the feedback. The WRF-Chem community provided a tool called "EPA_ANTHRO_EMIS" to create WRF-Chem compatible hourly anthropogenic emission input files from Sparse Matrix Operator Kernel (SMOKE) Modeling and mapped to the desired gas and aerosol chemistry mechanisms. The tool can be downloaded through the NCAR WRF-Chem tools. This tool will interpolate the 4 km grid spacing NEI data to any resolution one wishes to use for WRF-Chem simulations. Each WRF-Chem model grid point data is based on interpolation from NEI datasets. The method works well when WRF-Chem grid spacing is coarser than 4 km (our 9 km outer domain). With our inner domain at 3 km, it is possible to generate some unrealistic spatial patterns. However, we have found out that with a 3 km grid resolution, the emission representation is acceptable to use. If a higher resolution (i.e., 1 km) is desired, then more sophisticated interpolation is needed to overcome this issue. We have looked at the emission input files, and the tool aggregates the elevated point source emissions to the surface layer.

2. *It is advisable to introduce the data, data processing, and quality control before the weather/air quality description of the study area. How did you calculate the monthly wind directory? Average them simply may result in biases.*
   Thank you for the suggestion. We have rearranged the section by introducing the datasets first, and then the study area and model setup.

   For the wind direction, the plotted monthly data represent the average of daily max wind direction, which could cause biases in representing the prevailing winds. Therefore, we have recalculated the monthly wind direction by summing up north-south (V) and east-west (U) wind magnitude separately, and then calculated the direction (angle) by taking the arctan(U/V). We have also updated Figure 1 and added an explanation of the monthly wind direction.

*Minor:*

1. *T, RH: define them upon their first appearance.*

   Thank you for the comment. We have added the definitions in line 40 at their first appearance and replaced their full names with this definition throughout the paper.

2. *L317: reference for the BVOC contribution.*

Thank you for the suggestion. We have restated the sentence about the BVOC and added a reference from Guenther et al. 2012.

3. *L430: The PBLH in PHX shows the biggest negative bias (-509.7 m) while its O3 is underestimated. Are there any other reasons?*

   The underestimation of PHX $O_3$ is mainly during the daytime while nighttime $O_3$ is overestimated. Our hypothesis is that the underestimation of nightly PBLH contributes to the overestimate of $NO_2$ (Figure 8) and $O_3$ (Figure 3), which will cause the underestimate of $O_3$ during the daytime under a VOC-limited regime at PHX.

4. *Line 456: should be Figure 10*

   Revised accordingly.

5. *L556~580: reorganize.*

   Thank you for the feedback. We have removed the redundant sentences and reorganized the paragraphs.

6. *L638: O3, subscript*

   Revised accordingly.

Reviewer #3

*This work evaluates surface ozone simulated using WRF-Chem against several observational networks for three cities in Arizona. The analysis and paper itself are comprehensive. I recommend publication after several minor revisions specified below:*

*Page 7 line 184: Why use a 1 deg meteorological model for initial and boundary conditions when there are other higher resolution products for this too? Why not use a finer horizontal resolution meteorological model like NAM or HRRR? How might this impact your meteorological analysis later? This would in particular impact wind direction and speed. How did the wind speed and wind direction from AQS compare with the model results? You show the observations in Figure 1, but do not show how this compares with the model results.*

Thank you for the feedback. We agree that with higher spatial or temporal resolution products as initial and boundary conditions model performance might improve. Our model domain was initially designed to include Mexico as Arizona is at the boundary between US and Mexico, also in the summertime during the North American Monsoon winds shifted more southerly from Mexico which brings moisture, emissions, etc. We have looked at both NAM and HRRR. However, neither HRRR nor NAM extends to the southern part of Mexico. Right now, we are conducting another model-based analysis by using the tags to help understand the contribution of fire

plumes/smoke on urban air quality with cases. Under this analysis, we slightly modified the domain setup and have successfully run WRF-Chem with 12-km NAM.

We didn't compare the simulated wind speed and direction directly with AQS observations as our focus is more on the local $O_3$ production. However, the monthly mean wind fields presented in Figure 6f showed that the simulated prevailing wind in June is close to the observations in Figure 1g-1f.

*Page 8 line 213: Do you mean 30 days in June? If not, why not run for the entire June time period?*

Thank you for the comment. The model ran through the entire June, but for evaluation with AQS observations, considering the time difference between local time and UTC, we didn't count June 30th as an entire day.

*Page 9 line 217: For the selection of AQS sites can you describe this more? Did you only select sites with a certain amount of data available for the entire time window of 2017 – 2021? What latitude / longitude constraints did you use to define the region of a city? The AQS sites include metadata to characterize the site measurement scale, which gives you an idea of how representative that site is of a broader area. Some AQS sites are in locations that are less applicable to be simulated by a 3 km horizontal resolution model. Did you consider this at all in your choice of selected sites? And / or does this explain some of the biases you see in the results section?*

Thank you for your insightful comments regarding the selection of AQS sites for our study.

For the PHX metropolitan area, we initially focused on AQS sites with $O_3$ measurements within the non-attainment area (NAA) during the study period of 2017–2021. However, we recognize that even within the PHX metropolitan area or the NAA, there are variations in population density and emissions, as indicated by the spatial maps in Figures 10 and 13. To ensure representation across the central metro area of Phoenix, we selected 10 AQS sites for $O_3$ that are geographically dispersed. We acknowledge that with a 3 km horizontal resolution, multiple sites may fall within the same grid box. Additionally, individual AQS sites may exhibit variations in $O_3$ measurements, even when located in close proximity.

To address these challenges and ensure robust comparisons, we adopted a methodology recommended by our collaborators from ADEQ. Specifically, we compared the maximum hourly or maximum daily 8-hour average (MDA8) $O_3$ levels among all 10 selected sites with the maximum simulated $O_3$ concentration within the same area. This approach allows us to account for spatial variability and variations between individual sites while evaluating model performance.

*Page 11 section 3.1: I'm assuming from Section 2.3, that this is an average of many sites in a given city. Can you provide a table in the supplement for which sites you included in this analysis? Can you do this for other measurements if needed too? Is there variability in hourly ozone and MDA8 ozone across the different sites in a given city and how well does the model represent this variability?*

Thank you for the suggestion. We added a table listing the name, lat/lon, site number, and available measurements in Table S2 for reference. As mentioned above, the analysis of model evaluation to AQS was calculated by comparing the average of hourly or maximum daily 8-hour average (MDA8) $O_3$ levels among all selected sites with the mean simulated $O_3$ concentration within the same area. We have added the statement at the beginning of section 3.1 as well.

The variability does exist across different sites within the NAA, as can be seen from Figures 10c, 10f, 10i. Because of the limitation of model grid resolution and emission inventory resolution, it is challenging for the model to represent the hourly $O_3$ across sites. However, we have found that the model does have a better performance in simulating MDA8 $O_3$ as shown in Table 3, Figure 7&10.

*Table 1 and 2: Are these averaged for June 2017 – 2021 or just for a specific year.*

Thank you for the comment. We have updated the caption of Table 1 for better clarification. Tables 1 and 2 are the averages across June 2017-2021. The statistics for individual years for Table 2 (MDA8 $O_3$) are included in Table S3.

*Figure 5: Especially for isoprene and HCHO there do seem to be some differences between the model and observations. Can you add the median or mean bias to the plot to show this better?*

Thank you for the suggestion. We have added the mean bias (MB) for isoprene and HCHO in Figure 5.

*Figure 5: For Figure 5 and in plots/analysis later on, for NO2 is this a direct comparison with NO2 in the model to NO2 in the observations or do you apply any correction for interferences that the AQS sites can have for NO2 (e.g., Dunlea et al., 2007, https://doi.org/10.5194/acp-7-2691-2007). If no correction was applied, do you think this could explain some of the biases you see?*

Thank you for your valuable feedback regarding the AQS $NO_2$ measurements. In our study, we directly compare $NO_2$ concentrations derived from our model outputs with those observed at the Air Quality System (AQS) sites. We acknowledge the potential for interferences at AQS sites, as discussed by Dunlea et al. (2007), and we appreciate the reference to their work.

The traditional chemiluminescence FRM employed by AQS sites is subject to potential measurement biases resulting from interference by $NO_Z$ species. However, according to EPA (https://www.epa.gov/system/files/documents/2022-08/NO2_2021.pdf), within metropolitan areas, where a majority of the $NO_2$ monitoring network is located, $NO_2$ concentrations tend to be most heavily influenced by strong local $NO_x$ sources, thus the potential for $NO_Z$ related measurement bias is relatively small.

*Figure 6: Why not include all the different monitoring sites for these cities and zoom in more to show the regional variation across a city here rather than an average for all sites in a city? If this is an average, it would be good to be clearer here and in the text.*

Thank you for your valuable feedback and suggestion. Figure 6 serves to illustrate the spatial variation of mean $O_3$ and its precursors, providing insights into background pollutant levels and meteorological conditions across Arizona. We opted to present the average values across multiple monitoring sites for each specific city to offer a comprehensive overview. However, we acknowledge the importance of showcasing regional variations within cities. In subsequent sections when discussing $O_3$ exceedance over the Phoenix-Mesa non-attainment area, we zoom in and include data from multiple sites to provide a more detailed analysis.

*Figure 10: Including the WRF-Chem and reanalysis model simulated number of exceedance days corresponding with the AQS sites for direct comparison against the observations would be extremely useful for understanding the forecast skill of the WRF-chem model. It's hard to discern this from Figure 10. Can this be added to the analysis in some way? Your conclusions and Section 3.2 state that WRF-Chem agrees quite well with the observational data for number of exceedance days, but statistics or a bias plot would support this conclusion better.*

Thank you for the feedback. We have revised Figure 10 by incorporating AQS observations overlaid on WRF-Chem results to show a clearer comparison. It is obvious that variations exist between sites, even for adjacent ones within the same model grid. Therefore, conducting statistical analysis for individual site comparison poses challenges due to these variations. We believe the updated figure enhances the representation of model performance and provides better insights into the agreement between WRF-Chem simulations and observational data.

*Figure S5: Can you double check the units for ozone along the trajectory. The color bar goes from 0, 2, 4, 6, 8, 1? And the units are in ppm? This seems too high?*

Thank you for the feedback. The $O_3$ color bar of Figure S5 was cropped out and the unit was wrong. The color bar ranges are 0~100 ppb. We have updated the unit in the subplot title and adjusted the figure.

*Page 26 line 520 and also in your conclusion on page 31 line 618: Can you explain further why these results suggest that ozone exceedances on this day were caused by inter-state transport rather than local production? How far back in time do these HYSPLIT trajectories go? You also state that the PBLH was lower and temperature was higher which could cause higher local ozone formation too? Looking at the wind direction and speed from the model and observations seems important for evaluating this event. Have you looked at these metrics too? Adding this as a conclusion on page 31 and line 618 "that Arizona is substantially affected by inter-state transport of O3 from California" seems speculative. You need more analysis to state this, so I would strongly recommend rewording this sentence or doing more analysis.*

Thank you for your valuable insights. We have revised the speculative sentence in sections 3.3 and the conclusions regarding this event and updated the caption of Figure S5 to include the runtime of back trajectories.

As depicted in Figure 12, at the onset of the extreme events on June 13, 2017, temperatures were lower compared to previous days. However, as the event progressed, a heat episode emerged over Phoenix following a decrease in PBLH. The 48-hour back trajectories suggest a potential influence

of airmasses (O$_3$ or its precursors) originating from California or Asia contributing to the elevated O$_3$ levels observed in Phoenix on June 13, 2017. Subsequently, in the following days, the high O$_3$ concentrations are more associated with local production.

*Page 28 line 550: Can you provide the references for why you use these specific characteristics of FNR to describe the differences in the regimes? Additionally, there are several studies (e.g., Schroeder et al., 2017, https://doi.org/10.1002/2017JD026781) that demonstrate the uncertainties of using the FNR approach to approximate ozone production. Can you provide more context on the uncertainty of this approach? Are there other studies that have investigated the ozone production in your cities of interest in the recent past that you can also refer to? Do they agree with your conclusions using this FNR approach?*

Thank you for the suggestion. We have included more information about the references. The concentration of formaldehyde (HCHO) serves as an indicator for volatile organic compound (VOC) reactivity as it exhibits a positive correlation with proxy radicals (Sillman, 1995). Sillman (1995) identified that elevated HCHO/NOy ratios typically indicate NOx-limited regimes, whereas reduced HCHO/NOy ratios are indicative of VOC-limited regimes. Martin et al. (2004) found that during summer, the transition between radical- and NOx-limited regimes occurs at a particular ratio threshold. Using the Community Multiscale Air Quality (CMAQ) model with finer resolution for the entire continental U.S., Duncan et al. (2010) proposed that formaldehyde-to-nitrogen oxides ratios (FNRs) below 1 suggest a VOC sensitivity regime, FNRs between 1 and 2 indicate a transition zone between VOC and NOx sensitivities, and FNRs above 2 are characteristic of a NOx-sensitive regime. It is important to note that variations in meteorological variables, emission sources, and pollution levels can alter the ozone production regime. In different studies, various FNR thresholds are calculated. i.e., satellite column retrievals of FNR of 0.7–2.3 in Schroeder et al. (2017), and 3.2–4.1 in Jin et al. (2020). In addition, Acdan et al. (2022) used ground-based PAMS measurements and suggested a FNR of 0.3–1.0 for transition over the Lake Michigan region. In our study, we are following Duncan et al. (2010) which linked FNR with surface O$_3$ sensitivity in model simulation and used in several studies (Tang et al., 2012; Jin and Holloway, 2015, Souri et al., 2017) by defining FNRs less than 1 as VOC sensitivity regime, FNRs between 1 and 2 as a transition between VOC and NOx sensitivities ('the transitional zone'), and FNRs greater than 2 as NOx-sensitive regime. The definitions are updated in the text in section 3.4.

*Figure 14: Can you add lines to represent the different regime changes as specified in line 550 that you are assuming in this work? This would be useful further validation of these values.*

Thank you so much for your valuable suggestion. We have updated Figure 14 with lines of FNR equal to 1 and 2 to represent the different regimes.

*Page 31 line 624 - 630: Can you be clearer in this paragraph what your main conclusion is with regard to the ozone production sensitivity including the approach used to determine it and the uncertainties of this approach? From a policy perspective you state both your correlation approach strongly suggests a VOC-limited regime while also saying the FNR analysis suggests VOC-limited or transitional regime. Which regime does your analysis support and what is the uncertainty on it? It is important to be clear what your analysis is suggesting and the uncertainty on your analysis for understanding the policy ramifications of your work.*

Thank you for your suggestion. We have revised the paragraph in section 4 to clarify the statements regarding the ozone production regime.

The data presented in Figures 11 and 14 represent averages for the central Phoenix area and indicate a VOC-limited regime. Additionally, the spatial maps of FNR in Figure 13 highlight that 
[revised manuscript text omitted]

| City | AQS Site Number | Site Name | Latitude | Longitude | Measurements |
|------|-----------------|-----------|----------|-----------|--------------|
| Phoenix | 40130019 | West Phoenix | 33.48378 | -112.14256 | $O_3$, CO, $NO_x$ |
| Phoenix | 40139997 | JLG Supersite | 33.503833 | -112.095767 | $O_3$, CO, $NO_x$ |
| Phoenix | 40133002 | Central Phoenix | 33.45797 | -112.04659 | $O_3$, CO, $NO_x$ |
| Phoenix | 40134003 | South Phoenix | 33.40314 | -112.07526 | $O_3$, CO |
| Phoenix | 40133003 | South Scottsdale | 33.47968 | -111.91721 | $O_3$, CO, $NO_2$ |
| Phoenix | 40134005 | Tempe | 33.41123 | -111.93471 | $O_3$, CO |
| Phoenix | 40131003 | Mesa | 33.41018 | -111.86536 | $O_3$, CO |
| Phoenix | 40137022 | Lehi (Fire Station) | 33.474609 | -111.805769 | $O_3$ |
| Phoenix | 40137024 | Salt River High School | 33.508125 | -111.83852 | $O_3$ |
| Phoenix | 40137020 | Senior Center | 33.488131 | -111.855443 | $O_3$ |
| Tucson | 40191028 | Children's Park | 32.29515 | -110.9823 | $O_3$, CO, $NO_2$ |
| Tucson | 40191011 | 22nd and Craycroft | 32.204411 | -110.878067 | $O_3$, CO, $NO_2$ |
| Tucson | 40191032 | Rose Elementary | 32.172995 | -110.980134 | $O_3$ |
| Tucson | 40190021 | Saguaro National Park East | 32.174538 | -110.737116 | $O_3$ |
| Tucson | 40191021 | Cherry and Glenn | 34.403052 | -119.457914 | CO |
| Tucson | 40191034 | Coachline | 32.38082 | -111.12716 | $O_3$ |
| Tucson | 40191018 | Tangerine | 32.42526 | -111.064 | $O_3$ |
| Yuma | 40278011 | Yuma Supersite | 32.690278 | -114.61444 | $O_3$ |
| Yuma | 800268012 | San Luis Rio Colorado Well | 32.466389 | -114.768611 | $O_3$ |

140

**Table S3. Evaluation of MDA8 $O_3$ over Phoenix, Tucson, and Yuma for individual years for WRF-Chem simulations. The datasets for evaluation include AQS observations, CMAQ reanalysis (2017-2018), and ADEQ forecasts (2019-2021).**

145

| | | Phoenix | | | | | Tucson | | | | | Yuma | | | | |
|---|---|---|---|---|---|---|---|---|---|---|---|---|---|---|---|---|
| Year | | 2017 | 2018 | 2019 | 2020 | 2021 | 2017 | 2018 | 2019 | 2020 | 2021 | 2017 | 2018 | 2019 | 2020 | 2021 |
| AQS | | 67.9 | 66.6 | 64.1 | 62.0 | 67.8 | 60.1 | 54.9 | 54.8 | 54.6 | 57.2 | 54.3 | 53.8 | 56.1 | 48.2 | 52.1 |
| WRF Chem | | 66.8 | 62.6 | 58.9 | 59.2 | 62.6 | 60.2 | 55.3 | 52.8 | 54.3 | 56.8 | 60.5 | 61.8 | 58.5 | 57.8 | 57.7 |
| CMAQ /ADEQ | | 58.4 | 57.5 | 67.7 | 65.1 | 68.5 | 53.7 | 51.1 | 55.6 | 55.6 | 57.1 | 64.1 | 64.7 | 53.8 | 50.8 | 50.4 |
| R (W) | | 0.81 | 0.69 | 0.61 | 0.56 | 0.62 | 0.79 | 0.70 | 0.68 | 0.68 | 0.26 | 0.85 | 0.50 | 0.72 | 0.40 | 0.83 |
| R (C/A) | | 0.87 | 0.66 | 0.44 | 0.66 | 0.19 | 0.71 | 0.78 | 0.55 | 0.77 | 0.22 | 0.44 | 0.30 | 0.82 | 0.31 | 0.71 |
| MB (W) | | -1.09 | -3.94 | -5.21 | -2.76 | -5.16 | 0.14 | 0.37 | -1.98 | -0.35 | -0.40 | 6.16 | 7.95 | 2.32 | 9.62 | 5.62 |
| MB (C/A) | | -9.5 | -9.1 | 3.1 | 1.2 | 0.0 | -6.4 | -3.8 | 0.7 | -1.1 | -0.5 | 9.8 | 10.9 | -2.7 | 2.8 | 0.6 |
| ME (W) | | 6.3 | 6.6 | 7.0 | 6.6 | 8.1 | 4.7 | 4.7 | 4.3 | 4.8 | 7.0 | 9.1 | 11.8 | 5.5 | 10.1 | 6.8 |
| ME (C/A) | | 9.9 | 10.1 | 5.1 | 4.2 | 9.9 | 7.4 | 5.1 | 3.9 | 4.0 | 6.0 | 13.8 | 14.2 | 4.1 | 6.8 | 6.8 |
| RMSE (W) | | 7.6 | 8.5 | 8.3 | 8.3 | 10.4 | 5.7 | 5.9 | 4.9 | 6.2 | 9.0 | 11.9 | 15.5 | 6.6 | 13.2 | 9.5 |
| RMSE (C/A) | | 11.1 | 12.7 | 7.4 | 5.4 | 12.6 | 8.8 | 6.6 | 4.9 | 4.9 | 6.9 | 16.6 | 17.5 | 5.5 | 8.0 | 8.3 |
| NMB (W, 100%) | | -1.6 | -5.9 | -8.1 | -4.5 | -7.6 | 0.2 | 0.7 | -3.6 | -0.6 | -0.7 | 11.3 | 14.8 | 4.1 | 20.0 | 10.8 |
| NMB (C/A, 100%) | | -14.0 | -13.6 | 4.8 | 1.9 | -0.1 | -10.7 | -6.9 | 1.3 | -2.0 | -0.8 | 18.0 | 20.2 | -4.9 | 5.8 | 1.1 |
| NME (W, 100%) | | 9.2 | 10.0 | 10.8 | 10.7 | 12.0 | 7.8 | 8.6 | 7.8 | 8.7 | 12.3 | 16.8 | 22.0 | 9.8 | 21.0 | 13.0 |
| NME (C/A, 100%) | | 14.6 | 15.2 | 8.0 | 6.7 | 14.6 | 12.3 | 9.3 | 7.2 | 7.3 | 10.6 | 25.3 | 26.3 | 7.3 | 14.2 | 13.0 |
| MNB (W, 100%) | | -2.0 | -6.0 | -7.7 | -4.3 | -7.0 | 0.3 | 1.0 | -3.2 | 0.2 | 0.0 | 11.6 | 18.5 | 5.0 | 21.3 | 12.3 |
| MNB (C/A, 100%) | | -14.5 | -13.9 | 5.9 | 2.1 | 1.8 | -10.5 | -7.1 | 2.1 | -1.5 | 0.1 | 25.8 | 25.9 | -4.4 | 7.8 | 4.1 |
| MNE (W, 100%) | | 9.4 | 10.0 | 11.0 | 10.6 | 11.4 | 7.8 | 8.7 | 8.0 | 8.7 | 12.5 | 16.9 | 24.6 | 10.1 | 22.1 | 14.2 |
| MNE (C/A, 100%) | | 14.9 | 15.3 | 8.9 | 6.6 | 13.7 | 12.2 | 9.4 | 7.6 | 6.9 | 10.7 | 31.0 | 30.5 | 7.0 | 14.9 | 14.1 |
| FB (W, 100%) | | -2.7 | -6.9 | -8.6 | -5.3 | -8.0 | -0.2 | 0.4 | -3.6 | -0.4 | -1.3 | 9.5 | 14.2 | 4.3 | 17.9 | 10.5 |
| FB (C/A, 100%) | | -16.1 | -16.1 | 5.0 | 1.8 | 0.4 | -11.7 | -7.9 | 1.6 | -1.8 | -0.7 | 19.3 | 19.7 | -4.8 | 6.4 | 2.6 |
| FE (W, 100%) | | 9.6 | 10.7 | 11.7 | 11.3 | 12.2 | 7.7 | 8.8 | 8.1 | 8.7 | 12.7 | 15.3 | 21.0 | 9.7 | 18.8 | 12.4 |
| FE (C/A, 100%) | | 16.5 | 17.5 | 8.1 | 6.4 | 13.9 | 13.2 | 10.2 | 7.3 | 7.0 | 10.7 | 24.9 | 24.6 | 7.3 | 14.1 | 13.4 |

[Figure]

**Figure S1. June monthly mean O₃, CO, and NOₓ from WRF-Chem simulations for 2017 to 2021.**

1155

[Figure]

**Figure S2. Radiosonde observed and WRF-Chem simulated PBLH in June 2017.**

[Figure]

**Figure S3. Radiosonde observed and WRF-Chem simulated PBLH in June 2019.**

1160

[Figure]

**Figure S4. Radiosonde observed and WRF-Chem simulated PBLH in June 2020.**

[Figure]

**Figure S5. 48-hour HYSPLIT back trajectories for the observed O₃ exceedance event on 13 June 2017 in both (a) Phoenix and (c) Yuma, and the corresponding O₃ concentrations along the trajectories.**

1165